# Evaluating the Met Office Unified Model land surface temperature in Global Atmosphere/Land 3.1 (GA/L3.1), Global Atmosphere/Land 6.1 (GA/L6.1) and Limited Area 2.2km configurations.

Jennifer K. Brooke[1], R. Chawn Harlow[1], Russell L. Scott[2], Martin J. Best[1], John M. Edwards[1], Jean-Claude Thelen[1], Mark Weeks[1]

[1]Met Office, Fitzroy Road, Exeter, EX1 3PB, UK
[2]Southwest Watershed Research Center, USDA-ARS, 2000 E. Allen Road, Tucson, AZ 85719, USA

*Correspondence to*: J. K. Brooke (jennifer.brooke@metoffice.gov.uk)

**Abstract.** A limitation of the Met Office operational data assimilation scheme is that surface-sensitive infrared satellite sounding channels cannot be used during daytime periods where Numerical Weather Prediction (NWP) model background land surface temperature (LST) biases are greater than 2 K in magnitude. The Met Office Unified Model (UM) has a significant cold LST bias in semi-arid regions when compared with satellite observations, and a range of UM configurations were assessed with different model resolution, land surface cover datasets and bare soil parameterisations. UM LST biases were evaluated at global resolution and in a Limited Area Models (LAM) at 2.2 km resolution over the SALSTICE (Semi-Arid Land Surface Temperature and IASI Calibration Experiment) experimental domain in southeastern Arizona. This validation is in conjunction with eddy-covariance flux tower measurements. LST biases in the Global Atmosphere/Land 3.1 (GA/L3.1) configuration were largest in the mid-morning with respect to Moderate Resolution Imaging Spectroradiometer (MODIS) Terra (-13.6±2.8 K at the Kendall Grassland site). The diurnal cycle of LST in Global Atmosphere/Land 6.1 (GA/L6.1) showed a significant improvement relative to GA/L3.1 with the cold LST biases reduced to -1.4±2.7 K and -3.6±3.0 K for Terra and Aqua overpasses, respectively. The higher resolution LAM showed added value over the global configurations.

The spatial distribution of the LST biases relative to MODIS and the modelled bare soil cover fraction were found to be moderately correlated (0.61±0.08) during the daytime, which suggests that regions of cold LST bias are associated with low bare soil cover fraction. Coefficients of correlation with the shrub surface fractions followed the same trend as the bare soil cover fraction although with a less significant correlation (0.36±0.09), and indicate that the sparse vegetation canopies in southeastern Arizona are not well represented in UM ancillary datasets. The x-component of the orographic slope was positively correlated with the LST bias (0.41±0.05 for MODIS Aqua) and identified that regions of cold model LST bias are found on easterly slopes and regions of warm model LST bias are found on westerly slopes. An overestimate in the modelled turbulent heat and moisture fluxes at the eddy-covariance flux sites was found to be coincident with an underestimate in the ground heat flux.

# 1 Introduction

Infrared radiance data from hyperspectral satellite sounding spectrometers make up the largest proportion of assimilated data at the Met Office and over the last two decades have had the greatest forecast impact of any type of observation currently assimilated (English et al., 2000; Cardinali, 2009). The assimilation of a small selection of hyperspectral channels have been shown to improve estimates of temperature and humidity profiles for the initial state of NWP forecast (Hilton et al., 2012). However, a significant limitation of the assimilation scheme is that surface-sensitive hyperspectral channels cannot be used during daytime periods due to biases in the NWP model background land surface temperature (LST) and emissivity. The model background refers to a short-range model forecast; each data assimilation cycle uses newly received observations to update the model background in order to produce a model analysis (Rabier et al., 2005). At the Met Office, IASI (Infrared Atmospheric Sounding Interferometer) surface-sensitive channels, including window channels and lower-tropospheric (below 400 hPa) sounding channels, are rejected during assimilation windows for observations over land surfaces and during daytime periods (Pavelin and Candy, 2014).

Land surface temperature is the radiative skin temperature of the land and knowledge of the LST provides information on the temporal and spatial variations of the surface equilibrium state (Kerr et al., 2000). Recently, research trials have been completed at the Met Office which use night-time LST from the European Space Agency GlobTemperature LSTs project (Ghent et al., 2016) in the land data assimilation system; the study demonstrated improvements in near surface air temperature forecasts and soil temperatures (Candy et al., 2017). The required LST uncertainty for assimilation within the Met Office operational assimilation scheme is less than 2 K in magnitude, and Candy et al., (2017) highlights the large errors in daytime LST which must be overcome in order to further advance NWP data assimilation. Currently, as LSTs are not assimilated into the operational UM, they provide an independent source of data for assessing the performance of the land surface model's surface exchange and the boundary layer schemes (Edwards, 2010).

There are large systematic biases in the UM background land surface temperature which vary both spatially and temporally and they occur most strongly in semi-arid regions such as the south-west US, the Sahel, and south-central Asia. Land surface temperature biases in semi-arid regions are not limited to the UM and have been recognised as a source of model error in other land surface models (Guedj et al., 2011; Trigo et al., 2015; Zheng et al., 2012). Zheng et al., (2012) identified a 10 K cold bias over the western continental US in the Noah land model, and were able to successfully minimise the bias through a new formulation of the momentum and thermal roughness lengths, whilst Chen and Zhang (2009) found that the coupling strength in this model was too strong over short vegetated surfaces. Trigo et al., (2015) showed that in semi-arid areas the European Centre for Medium-Range Weather Forecasts (ECMWF) land surface scheme, HTESSEL (Hydrology Tiles ECMWF Scheme for Surface Exchanges over Land) underestimated in the daily amplitude of surface temperature. This has resulted in an overestimate of night-time LST (warm bias) and an underestimate in daytime temperatures (cold bias). Trigo et al., (2015) found that reducing the magnitude of the skin conductivity, which parameterises the thermal connection between

the surface and the soil by controlling the heat transfer to the ground by diffusion, led to a strengthening of the amplitude of the simulated diurnal cycle of surface temperature.

Near-surface air temperatures and LST are controlled by the surface energy balance (Prince et al., 1998). The warming of the land surface is forced by solar heating, and the dissipation of heat is partitioned between the sensible heat flux (H), the latent heat flux (LE), the ground heat flux (G) and the outgoing longwave radiation. The surface albedo describes the fraction of incident solar radiation reflected by a surface and is an important surface property in controlling the available energy. The correct partitioning of surface net radiation between the latent heat fluxes and sensible heat fluxes is critical (Oke, 1987; Rowntree, 1991; Dickinson, 1991) as this drives the diurnal development of the atmospheric boundary layer (Henderson-Sellers and Brown, 1993). The moisture content of the soil has a strong control over the partitioning of available energy between the heat fluxes (Castelli et al., 1999). In coupled models, land surface models (LSMs) provide the surface boundary conditions for atmospheric models, and therefore it is an important challenge in the development of LSMs to represent these processes that control the exchange of water and energy fluxes at the soil-atmosphere interface. The Joint UK Land Environment Simulator (JULES) (Best et al., 2011; Clark et al., 2011) is the land surface model that is coupled to the Met Office Unified Model (UM). Global scientific configurations of the land are identified as Global Land (GL) whilst the atmosphere is identified as Global Atmosphere (GA).

The Semi-Arid Land Surface Temperature and IASI Calibration Experiment (SALSTICE) was carried out during May 2013 in southeastern Arizona, in order to investigate the biases in the land surface temperatures (LST) forecast by the Met Office Unified Model (UM) in this region. Our study focuses on a small semi-arid region in southeastern Arizona for a domain of 31.25-32.25 °N and 69-71.5 °W. In this region collocated airborne observations and eddy-covariance flux tower measurements at sites based in the Walnut Gulch Experimental Watershed and Santa Rita Experimental Range have been made. The SALSTICE airborne campaign took place during 12 to 21 May 2013 with the timing of the airborne campaign designed to occur at the time of maximum LST biases in the UM. The campaign involved the UK Facility for Airborne Atmospheric Measurements (FAAM) BAe-146 aircraft which carried out five flights with the objective to diagnose the surface temperature errors within the UM. The outcomes of the airborne measurements will be presented in a future paper.

In this study, we consider the term *model bias* to be a model error which is systematic rather than random, and refer to the bias as being the model background-minus-observed (B-O), i.e. where a model, on average, under- or overestimates a quantity relative to an observed state. The study evaluates statistics of the model background-minus-observed (B-O) residuals for a range of UM model configurations. This study will characterise the spatial distribution and the magnitude of the UM land surface temperature biases in this region, in order to understand the mechanisms which give rise to the spatial distributions. We diagnose sources of model error using coincident MODIS retrievals and eddy-covariance flux tower measurements. This paper will evaluate changes to the magnitude of the LST bias for the month of May (the month of maximum LST bias) for a six year analysis period from 2013 to 2018, and will attribute observed trends to changes in a range of UM model configurations.

This article is arranged as follows: Section 2 provides a description of the eddy-covariance sites and the instrumentation deployed, and the MODIS retrievals utilised. The UM configurations used in this evaluation are summarised. Results are presented in Section 3, including an assessment of the diurnal cycle of LST, an evaluation of LST biases for the UM configurations for different land classification types, and an examination of correlations between the spatial distribution of LST biases with modelled orography and surface fractional cover. An evaluation of the surface energy balance for the coupled UM configurations is presented. Section 4 then presents the conclusions.

## 2 Methodology

### 2.1 Eddy-covariance flux tower measurements

Eddy-covariance measurements offer model verification of the surface exchange processes, and provide an opportunity to examine sources of model error by investigating components of the surface energy balance (SEB) in the Unified Model. The model is evaluated at four eddy-covariance flux tower sites: Lucky Hills and Kendall Grassland, located in the USDA-ARS's Walnut Gulch Experimental Watershed, and the Santa Rita Grassland and Santa Rita Mesquite sites, located in the Santa Rita Experimental Range (Scott et al., 2015), all located in southeastern Arizona. The SEB and LST has been investigated during the period of 12 to 21 May 2013, coincident with the SALSTICE campaign. *In situ* measurements of LST from an infrared radiometer at the flux tower sites have been further evaluated for the period of 1 to 31 May 2014-2018. The study will evaluate surface temperatures for a six-year analysis period.

Lucky Hills Shrubland (Ameriflux site id: US-Whs) is a site dominated by Chihuahuan desert shrubs and is defined as open shrubland according to the International Geosphere-Biosphere Programme's (IGBP) land cover classification. Kendall Grassland (US-Wkg) and Santa Rita Grassland (US-SRG) sites both have perennial bunch grasses as their dominant vegetation and are assigned by IGBP as a semi-arid warm season desert grassland, and Santa Rita Mesquite (US-SRM) is a woody savannah site (IGBP) predominantly vegetated with small mesquite trees and grasses. These semi-arid ecosystems have bare soil cover in the range of 45 % (Santa Rita Grassland) to 63 % (Lucky Hills Shrubland) (Scott et al., 2015).

The data collected at these sites include screen-level air temperature, humidity, winds, long- and shortwave broadband hemispherical irradiances, sensible and latent heat fluxes, ground heat fluxes, soil temperature, rainfall, and land surface temperature (at Lucky Hills and Kendall Grassland). Details of instrumentation and a full description of the eddy-covariance flux tower sites can be found in Scott et al., (2015). Section 2.2.1 – 2.2.2 will briefly describe the corrections applied to the observational datasets pertinent to the evaluation presented in this study.

### 2.2.1 Corrections applied to the eddy-covariance measurements

Eddy-covariance techniques use measurements of vertical velocity fluctuations and scalar concentration fluctuations to produce a direct estimate of the vertical flux of sensible heat ($H_{meas}$) and latent heat ($LE_{meas}$). It is well established in the literature that there is difficulty in closing the SEB with eddy-covariance measurements associated with underestimates

in measured turbulent heat fluxes (Twine et al., 2000; Wilson et al., 2002; Foken et al., 2008). Wilson et al., (2002) have shown that these errors can account for 10 to 30 % of the net radiation, and for the eddy-covariance flux tower sites used in this study, Scott et al., (2010) found that the energy balance errors for the 30-minute time averaging window account for 17-27 %. It is not expected to be able achieve an instantaneous energy balance closure at every time step due to the vegetation canopy heat storage. However, the canopy storage in the sparse canopies of southeastern Arizona is generally neglected as has been done in the methodology applied here.

The near surface ground heat flux measurements are at a depth of 5 cm from the surface soil layer, and subsequently a fraction of the surface soil heat flux is not measured. The correction methodology of Scott et al., (2009) has been applied to the ground heat flux data to account for the missing proportion of the soil heat flux. Additionally, soil heat flux plates buried in the soil can introduce measurement biases due to difference in conductivity between the measurement plates and the surrounding soil (Gentine et al., 2012). Finally, the ground heat fluxes are point measurements and as such do not represent the variability of fluxes across the fetch/sensing area in the same manner associated with the eddy-covariance measurements.

The use of SEB measurements in order to attribute model biases requires the conservation of energy to be achieved. In our study we assume the sole error is due to under sampling of the turbulent fluxes by the eddy-covariance measurements; and forces closure of SEB whilst maintaining the Bowen ratio ($BR$) (Twine et al., 2000). The Bowen ratio is the ratio of the sensible heat flux to the latent heat flux. In this method it is assumed that the measured ground heat flux ($G_{meas}$) is well measured and the corrected turbulent heat fluxes ($H_{corr}$ and $LE_{corr}$) represent closure of the surface energy balance.

**2.2.2 Corrections applied to the IRT surface temperature measurements**

An Apogee infrared radiometer (Bugbee et al., 1998), or IRT, installed at Lucky Hills and Kendall Grassland measures the upwelling longwave radiance across a spectral range of 8-14 µm. An estimate of the surface temperature can be made through a conversion of the measured upwelling longwave radiance using the Stefan-Boltzmann law and using an assumed surface emissivity of 1.0 (Fiebrich et al., 2003). The broadband emissivity of bare soil can vary substantially with values in the range of 0.81-0.99 (Ogawa et al., 2003). A correction is made to the measured upwelling longwave radiance, in order to account for such uncertainty in the surface emissivity, as described below.

The National Land Cover Database (NLCD) 2006 (Fry et al., 2011) has been used to identify shrubland and grassland regions of the SALSTICE airborne flight tracks (described in a future paper). The NLCD 2006 dataset is a 16-class land cover classification scheme that has been applied consistently across the United States at a spatial resolution of 30 meters. Emissivity retrievals from the airborne ARIES (Airborne Research Interferometer Evaluation System) instrument (Newman et al., 2005) have been performed from the SALSTICE campaign (not shown). An 8-14 µm broadband emissivity has been calculated for the surface types (shrubland and grassland) found at Kendall Grassland and Lucky Hills. The 8-14 µm

broadband emissivity was found to be 0.97±0.02. The variability in emissivity obtained from the ARIES measurements was found to have a ±1.1 K uncertainty on the land surface temperature from the daytime IRT measurements.

A further correction is applied which accounts for the downwelling longwave radiation according to Eq. (1).

$$BT_{surf,8-14\,\mu m} = \frac{1}{\varepsilon}\left(LW^{\uparrow}_{surf,8-14\,\mu m} - (1-\varepsilon)LW^{\downarrow}_{surf,8-14\,\mu m}\right) \qquad (1)$$

where $BT_{surf,8-14\,\mu m}$ is the surface blackbody radiance, $\varepsilon$ is the emissivity in the range of $0.97 \pm 0.02$, $LW^{\uparrow}_{surf,8-14\,\mu m}$ is the
upwelling radiance at the surface in the IRT field of view, $LW^{\downarrow}_{surf,8-14\,\mu m}$ is the downwelling radiance at the surface which is reflected into the IRT field of view.

The 8-14 µm downwelling longwave ($LW^{\downarrow}_{surf,8-14}$) is modelled using the Havemann-Taylor Fast Radiative Transfer Code (HT-FRTC) (Havemann, 2006) for each of the ground sites, Lucky Hills and Kendall Grassland, which have an IRT installed. Hourly downwelling longwave radiation is calculated based on the ECMWF ERA-Interim (Dee
et al., 2011) which is available every 6 hours (00, 06, 12 and 18). For the other times the ECMWF ERA-Interim atmospheric profiles have been interpolated in time. The downwelling calculation uses the 8-14 µm spectral emissivity for sandy soil from Arizona from UCSB (University of California, Santa Barbara) Emissivity Library (UCSB Library) (https://icess.eri.ucsb.edu/modis/EMIS/html/em.html). The IRT measurements were found to be on average (of the six years) -0.51 K colder when accounting for the reflected downwelling average for the 6 years; the smallest impact was found
for the 2014 measurements (-0.43 K) and the largest impact was found in 2015 (-0.59 K).

Cloud screening of the IRT data has been performed using coincident observations of downwelling shortwave as no direct measurement of cloud cover is made at the two AmeriFlux sites. The theoretical clear skies downwelling shortwave for each site has been calculated and compared with the measured downwelling shortwave; times where there is a suppression in the observed downwelling shortwave compared with the theoretical calculation has been attributed to the presence of cloud. It
was found that on average (for both sites and for the six analysis years) the IRT data was 0.45 K warmer when applying cloud screening which equates to a -0.45 K larger cold model bias. Cloud screening of the IRT data had a smaller impact in May 2013 and May 2018 with a -0.2 K colder model bias when compared with not accounting for cloud, and the largest impact was found for May 2015 and May 2016 contributing to a -0.7 K colder model bias.

The IRT measurements are only presented for daylight hours from 6 am to 6 pm local solar time. The IRT measurements
outside of this timeframe were anomalously warm and were identified as being unreliable. The advantage of these measurements is that they give greater diurnal variation at each site recorded at 30-minute intervals and compliment MODIS LST retrievals which have only four overpasses per diurnal cycle.

## 2.3 MODIS LST retrievals

The MOD11_L2 and MYD11_L2 LST products are generated using Moderate Resolution Imaging Spectroradiometer
(MODIS) radiances at 1 km spatial resolution and are comparable in the resolution with the 2.2 km LAM. Retrievals from

both Terra (10-11 am/pm overpass time, MOD11_L2) and Aqua (1-2 am/pm overpass time, MYD11_L2) of LST for May 2013-2018 are utilised. The LST retrieval from the Aqua platform is likely to be closer to the maximum daily LST than that acquired from the Terra platform (Coops et al. 2007).

The MODIS LST retrieval algorithm is described in the MODIS Land-Surface Temperature Algorithm Theoretical Basis Document (Wan & Dozier, 1996; Wan, 1999). In the literature it is found that the Collection 5 (C5) LST product has an accuracy to within 1-2 K (Coll et al., 2005; Wang et al., 2007; Wan et al., 2004). More recent studies have shown that the C5 retrievals underestimate LST by more than 3 K for particular bare soil/sand sites; the MODIS Collection 6 (C6) retrieval was developed to address these biases (Wan, 2014). For this reason we use both C5 and C6 products in our land surface temperature evaluation.

In order to produce an LST retrieval for each eddy-covariance flux site, boundaries of constant latitude and longitude were chosen such that the boundaries have tangent points 1 km away from each ground site. Thus, 2 km by 2 km boxes are formed about each site. MODIS pixels whose centres fall within these boxes are selected and averaged to give site-specific LST. Therefore, the number of MODIS pixels contributing to one of these site specific values can range from 1 to 5 dependent on where the site is within the swath of the instrument. Cloud screening of the MODIS data has been applied; data which was flagged by the MODIS quality algorithm as contaminated by cloud has been removed from the analysis.

Li et al., (2013) found that the difference in the LST measured in nadir and off-nadir satellite observations can be as large as 5 K for bare soils, and Hu et al. (2014) found that LST with smaller view angles tend to be warmer. Our results support this finding; we find a larger model cold LST biases when considering smaller view angles. Our analysis finds that the average LST bias with respect to Terra (Aqua) was 0.2 K (0.3 K) warmer at 40° relative to 30°; 0.6 K (0.8 K) at 45° relative to 30°; and 1.2 K (0.88 K) at 50° relative to 30°. The angular dependence described arises due to different viewing and illumination geometry of the surface; studies have shown that factors including slope orientation relative to sun, properties of the soil and vegetation such as the heterogeneity and the structure of the vegetation canopy, all contribute to the directional anisotropy (Duffour et al., 2016; Ermida et al., 2014; Rasmussen et al. 2010). Hence, overpasses were only included in the analysis if the incidence angle over the mid-point of the study area was less than 30°.

**2.4 Unified Model configurations**

The relevant configurations of the UM assessed in this paper are summarized in Table 1, which describes model changes between configurations including dynamics, resolution, data assimilation (DA) bias correction, initialisation, land cover and bare soil parametrisations. The operational models at the Met Office are continually monitored and developed in order to minimise systematic model biases and to improve forecasts. The changes in all model configurations evaluated in this study are part of the operational model development cycle. Understanding how the model configuration changes impact on surface temperatures in the development cycle, for the purpose of assessing where any advances in the assimilation of greater volumes of hyperspectral satellite sounding data, is an important evaluation. The UM configurations referred to in this study

are a coupled configuration consisting of specific configurations the UM atmospheric model (GAx.y) and the JULES land surface model (GLx.y).

The global configuration, GA/L3.1, was run at 25 km resolution with 70 vertical levels and used the New Dynamics dynamical core to solve the atmosphere's equations of motion (Davis et al., 2005; Walters et al., 2011). The operational GA/L6.1 configuration, introduced in 2015, used the ENDGame dynamical core to solve the atmosphere's equations of motion and used an increased horizontal resolution of 17 km, hereafter referred to as GA/L6.1_17km (Walters et al., 2017). The horizontal resolution of GA/L6.1 was further increased to 10 km (hereafter referred to as GA/L6.1_10km) which applies to the analysis of May 2018. The vertical resolution remained unchanged for all configurations. The GA/L3.1 configuration outputs three hourly diagnostics, and all GA/L6.1 configurations output diagnostics on an hourly basis. The analysis presented in this paper does not use of the first 7 hours of each forecast for all model configurations, as the first 3-6 hours of a forecast are generally regarded as not reliable because of the model spin-up time (Kasahara et al., 1992).

Bias correcting actively assimilated sounding radiance observations is necessary in order to generate an unbiased forecast analysis (Zhu et al., 2014). The global model used a static bias correction scheme (Harris and Kelly, 2001) in 2013-2015 whilst variational bias correction (VarBC) was introduced from 2016 onwards (Cameron and Bell, 2018). Global model configurations with the _static and _VarBC indicate the bias correction scheme used. The two schemes treat radiance observations differently, for example, in the static scheme bias corrections are pre-computed for all available sensors and the bias correction is typically updated at 6-12 month intervals. The bias corrections are based on an observation corrected to the model background (background field from previous model run). VarBC, in contrast, is an adaptive bias correction scheme, and the bias for each radiance channel is computed using a linear predictor model. The observations are corrected to the model analysis (rather than the background) given from the 4D-Var assimilation system. ASCAT volumetric surface soil moisture data is assimilated into all global configurations (Dharssi et al., 2011).

The nesting of high resolution LAMs provide useful information at scales that cannot be provided by lower-resolution global-scale models (Davis, 2014), for example from surface properties, such as orography and vegetation cover, and by better resolving moist physical processes (e.g. clouds, precipitation, visibility). Two operational nested LAMs were run for the contiguous US as part of the National Oceanographic and Atmospheric Administration's Hazardous Weather Testbed at 4.4 km and 2.2 km resolutions (referred to as US4.4 and US2.2 hereafter) (Hanley et al., 2016). The US4.4 was based on the European 4 km model (EURO4) and the US2.2 was based on the UKV (variable resolution UK model for kilometre scale forecasting) operational model. The US4.4 was initialized from the GA/L3.1 T+0 analyses and driven by hourly GA/L3.1 lateral boundary conditions. US2.2_ConfigA-B were nested within the US4.4 and initialized from the US4.4 T+3 forecast conditions and driven by hourly US4.4 lateral boundary conditions. There was no additional data assimilation in the LAMs. No further configurations of the US4.4 were run beyond 2014, and for this reason the US4.4 is not fully evaluated in this study. The US2.2 (ConfigC-E, 2015-2018) was initialised directly from the GA/L6.1 T+0 analyses and driven by hourly GA/L6.1 lateral boundary conditions. Specifically, US2.2_ConfigC was initialised from GA/L6.1_17km_static T+0; US2.2_ConfigD was initialised from GA/L6.1_17km_VarBC T+0; and US2.2_ConfigE was initialised from GA/L6.1_10km_VarBC T+0.

All global configurations and US2.2_ConfigA-D use the International Geosphere-Biosphere Programme's (IGBP) land cover classification dataset for the surface fractional cover mapped to JULES five Plant Functional Types (PFTs). US2.2_ConfigE uses the surface fractional cover based on the European Space Agency's Land Cover Climate Change Initiative (ESA LC_CCI) global vegetation distribution (Poulter et al., 2015; Harper et al., 2016), mapped to JULES five PFTs.

A tiled approach is used to represent sub-grid scale heterogeneity (Essery et al., 2003); the surface of each land point is subdivided into five types of vegetation, known as PFTs (broadleaf trees, needleleaf trees, temperate C3 grass, tropical C4 grass and shrubs) and four non-vegetated surface types (urban areas, inland water, bare soil and land ice). Surface exchange on these nine surface tiles can be calculated in two ways; either on each tile separately or by aggregating the surface properties on a single tile representing a grid-box mean. The global configurations amalgamate the properties of each surface tile, weighted by their grid-box fraction, into a single representative parameter value. As such there was no representation of sub-

grid heterogeneity (Walters et al., 2011). In contrast to this, the fluxes between the land surface and the atmosphere were calculated on each of the 9 surface tiles independently for the US2.2.

       A series of land surface parameters were varied between UM configurations as part of the operational implementation in order to improve the representation of near-surface temperature gradients and surface fluxes. These land surface parameters

are summarised in Table 1. In GA/L3.1 and US2.2_ConfigA the surface emissivity was set to 0.97 over all land surface tiles, however this was seen to cool the surface too strongly in desert regions (Walters et al., 2017). In all GA/L6.1 configurations and US2.2_ConfigB-E individual surface tiles have been assigned different emissivity parameter values; bare soil uses an emissivity of 0.90, and C3 grasses, C4 grasses and shrubs use an emissivity of 0.98. To summarise the emissivity changes, an emissivity map of the study region for each configuration is presented in Supplement Fig. S1. The emissivity changes relative

to GA/L3.1 (Fig. S1a) and US2.2_ConfigA (Fig. S1d) result in regional decreases for GA/L6.1 (Fig. S1b, Fig. S1c) and US2.2ConfigA-D (Fig. S1e) associated with regions of larger bare soil fractions. US2.2ConfigE (Fig. S1f), in contrast, shows an increase in emissivity for the study domain related to a reduction in the bare soil cover fraction. Section 3.3 provides a more thorough discussion of the surface heterogeneity and land cover in each model configuration.

       Surface exchange is treated using Monin and Obukhov (1954) mean similarity theory. The roughness length of

25 heat ($z_{OH}$) is required to estimate the sensible heat flux and can be considered relative to that of momentum ($z_{OM}$) through the simple ratio of $z_{OM}/z_{OH}$. GA/L3.1 uses a bare soil roughness length ($z_{OM}$) of 0.0032 m, and the ratio of roughness lengths for heat and momentum, $z_{OH}/z_{OM}$, was set to 0.1 for all land surface types. In all GA/L6.1 configurations (17km_static, 17km_VarBC and 10km_VarBC) and all configurations of the US2.2, the bare soil roughness length was reduced to 0.001 m and the ratio $z_{OH}/z_{OM}$ was treated independently for each surface type; the bare soil $z_{OH}/z_{OM}$ was decreased to 0.02 (Walters

et al., 2014; Walters et al., 2017). The $z_{OH}/z_{OM}$ ratio was revised between GA/L3.1 and GA/L6.1 in order improve both land surface temperature and near surface air temperatures in desert regions. The revised $z_{OH}/z_{OM}$ ratio was adopted in the US2.2 (and other LAMs) from 2013, whilst GA/L6.1 was adopted for operational use in July 2014.

       Model cloud-clearing has been performed for all model configurations based on a threshold of total cloud fraction greater than 0.1 for each model grid box. In cases where the combination of model and MODIS cloud clearing resulted in a

fraction of the domain contained less than 10 % of data, the comparison was excluded from the analysis as this was taken to indicate cloud in the region that could affect the measurements.

The surface temperature biases (observed–minus-model background, O-B) for the southern part of the North American continent are presented in Figure 1 for IASI 1D-VAR retrievals compared with two UM global configurations, GA/L3.1 (May 2013) and GA/L6.1_17km_static (May 2015). The IASI 1D-Var retrievals have a spatial resolution of 11 km and have been regridded to a half degree global resolution. In terms of model background-observations (B-O) surface temperature biases, it can be seen that GA/L3.1-IASI 1D-VAR gives rise to an east-west spatial divide in the magnitude of LST biases with LST cold biases in excess of -10 K in the south-west US, western Mexico and extend east into the Great Plains. Moderate cold LST biases extend into the northern US with biases in the range of -4 to -6 K. The North American mean bias is reduced in GA/L6.1_17km_static-IASI 1D-VAR compared with GA/L3.1-IASI 1D-VAR, although regional biases such as the south-west US are still prominent.

## 3 Results and discussion

### 3.1 Representation of the diurnal cycles of LST

The model diurnal cycles in surface temperature for Kendall Grassland are compared in Figure 2 against observations. The GA/L3.1 diurnal cycle (Figure 2a) highlight a cold model prediction when compared with MODIS retrievals; daytime biases range from $-13.6\pm2.8$ K and $-8.8\pm2.5$ K for Terra and Aqua overpasses, respectively. Biases in modelled LST are larger in the mid-morning associated with the Terra overpass which indicates the model struggles to capture the magnitude of the warming from the morning transition to the late morning period. Observations from Aqua are made approximately at the time of the maximum LST when surface temperatures are changing less rapidly than at the time of Terra observations. The biases seen with MODIS are consistent when comparing with measurements from the IRT (bias of $-7.5\pm3.2$ K).

The US2.2_ConfigA diurnal cycle (Figure 2b) shows that the phase of the surface temperature is improved relative to GA/L3.1. The US2.2_ConfigA configuration improves the timing of the initial warming during the morning transition, and the bias relative to Terra ($7.6\pm2.4$K) is improved as a consequence. The underestimate at the time of the diurnal maximum remains in the US2.2_ConfigA and the magnitude of the cold bias is approximately equal to GA/L3.1.

The diurnal cycle of surface temperature in GA/L6.1_17km_static (Figure 2c) shows a significant improvement relative to GA/L3.1. The cold LST biases is reduced to $-1.4\pm2.7$ K and $-3.6\pm3.0$ K for Terra and Aqua overpasses, respectively. There is additionally an improved overlap of the one sigma confidence intervals for the daytime LST measured by the ground-based IRT and for GA/L6.1_17km_static. The US2.2_ConfigC (Figure 2d) has a further small improvement of the LST bias relative to GA/L6.1_17km_static configuration, although not to the same extent as was seen between GA/L3.1 and the US2.2_ConfigA. Biases in US2.2_ConfigA are reduced to $-1.3\pm2.1$ K (w.r.t Terra) and $-2.5\pm1.6$ K (w.r.t. Aqua).

The LST measured at the ground sites are from Apogee IRT radiometers installed at 4 m and have a field of view which covers approximately 9 m$^2$. The model grid squares that contain these sites are large and in the case of the GA/L3.1 and

GA/L6.1_17km_static cover large elevation ranges within one grid square. As the model configurations have grid squares that are many orders of magnitude larger than this, the IRT-measured LST greatly under sample the variability within the model grid square, however despite this Figure 2c and Figure 2d demonstrate good agreement in the representation of the daytime diurnal cycle.

## 3.2 Evaluation of UM surface temperatures at eddy-covariance sites

This section extends the analysis to the four eddy-covariance sites, evaluates surface temperatures for different land classification types and will attribute observed trends to changes in a range of UM model configurations. Figure 3 presents the daytime LST biases for the UM configurations relative to MODIS C6 Terra and Aqua retrievals for the six years in the analysis (2013-18, row 1-6).

The US2.2_ConfigA-D have a smaller cold surface temperature biases compared with the corresponding global configuration from 2013-2017. The higher resolution US2.2 generally has a smaller daytime bias than the US4.4 (approximately 1 K smaller, data not shown). The US2.2 configurations have higher resolution ancillary datasets which better resolved surface properties, such as orography and surface fractional cover, and subsequently improve the model representation of the surface heterogeneity, than can be represented in GA/L3.1 and GA/L6.1 configurations. In addition, there is a reduction in the bare soil roughness length parameterisation (Table 1) in the US2.2_ConfigA ($z_{OM}$=0.0010 m and $z_{OH}/z_{OM}$=0.02) compared with GA/L3.1 ($z_{OM}$=0.0032 m and $z_{OH}/z_{OM}$=0.10) which is required to estimate the sensible heat flux. A smaller roughness length for heat results in a smaller sensible heat flux, and hence a smaller heat flux from the land surface to the atmosphere.

Improvements in LST biases in the US2.2, compared with GA/L3.1, are greater at the shrubland sites, Lucky Hills and Santa Rita Mesquite compared with the grassland sites, Kendall Grassland and Santa Rita Grassland. At Lucky Hills, for example, biases with respect to Aqua are reduced from -8.2±2.5 K (GA/L3.1, 2013) to -3.8±1.9K (US2.2_ConfigA). In contrast, at Santa Rita Grassland, the biases are reduced to a lesser extent from -10.7±3.4 K (GA/L3.1, 2013) to -7.3±1.7K (US2.2_ConfigA), and at Kendall Grassland the bias w.r.t Aqua is unchanged between GA/L3.1 and US2.2_ConfigA. The IRT measurements support this trend; at Lucky Hills the bias in reduced from -9.0±3.7 K (GA/L3.1) to -3.3+2.3 K (US2.2_ConfigA), whilst the IRT measurements at Kendall Grassland only show a 2.2 K improvement in the US2.2_ConfigA compared with GA/L3.1.

The higher resolution ancillaries in the US2.2 improve the surface fractions for the two shrubland sites; the US2.2 increases the bare soil fractional cover which acts to increase the sparsity of the vegetation cover, and improves the model representation of the surface heterogeneity. At the Lucky Hills shrubland site, for example, the bare soil fraction is increased from 0.26 (GA/L3.1) to 0.48 (US2.2_ConfigA-D) and at Santa Rita Mesquite a similar increase from 0.22 (GA/L3.1) to 0.37 (US2.2_ConfigA-D) is reflected. This brings the modelled bare soil cover fractions closer to the observed fractions of 63 % for Lucky Hills Shrubland and 50 % for Santa Rita Mesquite (Scott et al., 2015). However, at the two grassland sites, Kendall Grassland and Santa Rita Grassland, there was a reduction in bare soil fractional cover between GA/L3.1 and US2.2_ConfigA.

The lower cover fraction at the grassland sites is maintained in all GA/L6.1_17km configurations. At the Kendall Grassland site, for example, the bare soil fraction is decreased from 0.26 (GA/L3.1) to 0.20 (US2.2_ConfigA-D) and at Santa Rita Grassland a similar decrease from 0.16 (GA/L3.1) to 0.10 (US2.2_ConfigA-D) is reflected. This is in contrast with the observed fractions of 60 % for Kendall Grassland and 45 % for Santa Rita Grassland (Scott et al., 2015).

The trend observed suggests that for the shrubland sites land surface warming can be attributed to both the revised bare soil roughness lengths and increased fraction of bare soil, whilst at the grassland sites a decrease in bare soil fractional cover appears to have a cooling affect that offsets the warming associated with the updated roughness length parameterisation.

In US2.2_ConfigB, the Lucky Hills site is seen to warm too strongly compared with the three other eddy-covariance sites. The bare soil emissivity was reduced to 0.90 in US2.2_ConfigB, which acts to reduce the upwelling longwave radiation at the surface and leads to warming of surface temperatures at all four sites. At Lucky Hills, a warm surface temperature bias develops with respect to Terra C6 (4.6$\pm$4.5 K in 2014) and Aqua C6 (1.5$\pm$2.6 K in 2014). The IRT measurements located at Lucky Hills support the development of the warm bias (0.6$\pm$5.4 K in 2013; 1.4$\pm$2.6 K in 2015). Lucky Hills has the largest bare soil fraction of the four eddy-covariance sites, and therefore a greater change as a result of the revised bare soil emissivity is expected. Although too much warming is seen at Lucky Hills, the revised emissivity leads to improvements in the surface temperature bias at the other three eddy-covariance sites.

The GA/L6.1 and US2.2 configurations use the same set of bare soil parameters (same emissivity, $z_{OH}/z_{OM}$ and $z_{OM}$) and hence the main difference between configurations from the land perspective is the resolution of the configuration. In GA/L6.1_17km_static (2015; Figure 3, row 3), the warming of the land surface that was seen in the US2.2_ConfigB, is reflected in the global configuration. LST biases in GA/L6.1_17km_static at the two shrubland sites are reduced by 8-9 K with respect to Terra, and 3-5 K with respect to Aqua compared with GA/L3.1. The IRT measurements support the improved LST biases between the two global configurations. For example, at Lucky Hills, a reduction in the model bias from -9.0+3.7 K (GA/L3.1, 2013) to -2.7$\pm$2.46 K (GA/L6.1_17km_static, 2015) was found with respect to the IRT. The same trend is observed for Kendall Grassland; the bias is reduced from -7.5$\pm$3.2 K (GA/L3.1, 2013) to 0.15$\pm$2.4 K (GA/L6.1_17km_static, 2015). The LST bias in all GA/L6.1 configurations is generally smaller with respect to Terra than with respect to Aqua, whilst the reverse was true for GA/L3.1. This trend supports the improved phase of the LST diurnal cycle described previously.

The biases are generally larger in GA/L6.1_17km_VarBC/US2.2_ConfigD (2016, 2017) than in GA/L6.1_17km_static /US2.2_ConfigC (2015). This step change is coincident with a change in the bias correction scheme for satellite radiances from a static scheme to VarBC, between 2015 (GA/L6.1_17km_static) and 2016 (GA/L6.1_17km_VarBC). It could be expected that a change to the treatment of the bias correction could result in a different model climatology which consequently influences the magnitude of the surface temperature bias as was found for the model humidity field (Cameron and Bell, 2018). The magnitude of the biases with GA/L6.1_17km_VarBC are still improved compared with GA/L3.1_25km_static, even though there appears to be a degradation when compared with GA/L6.1_17km_static.

In 2018 it can be seen that the US2.2_ConfigE has a larger cold LST biases compared with GA/L6.1_10km_VarBC, and it is the only year in our analysis where the global configuration out-performs the higher resolution US2.2 configuration.

The GA/L6.1_10km_VarBC global configuration has an upgraded horizontal resolution of 10 km, and this exhibits an increase in the resolution of the surface fractional cover land surface ancillary. At all four eddy-covariance sites there is an increase in the shrub and bare soil cover fractions, and an associated decrease in the total grass fraction, and again this acts to increase the sparsity of the vegetation cover, and hence improves the model representation of the surface heterogeneity.

US2.2_ConfigE uses the ESA LC_CCI surface fractional cover dataset rather than the IGBP surface fractional cover dataset, and the trend observed suggests there is a degradation in the land surface temperature bias at all four sites relative to US2.2_ConfigA-D. The mechanism for the poorer performance will be discussed more fully in the following section.

        Of the four eddy-covariance sites evaluated in this study, the least improvement is seen for the Santa Rita Grassland site across the six year analysis period. At Santa Rita Grassland the model LST biases with respect to Aqua is generally greater

than 4-5 K for all configurations. In the experimental domain in southeastern Arizona, the dominant vegetation type is shrubland, and for this reason it could be expected that the land surface ancillaries for the grassland sites, despite any differences in model resolution, are not as well represented as for the shrubland sites. The Santa Rita Mesquite site, in contrast, has surface temperatures biases which are below 2 K during 2015 and 2016, and could therefore be considered small enough to be suitable for data assimilation purposes.

As discussed in the methodology, it has been shown in the literature that MODIS C5 retrievals underestimate LST by more than 3 K for particular bare soil/sand sites (Wan 2014), and therefore it was important to evaluate MODIS C5 and MODIS C6 in order to access the impact on the magnitude of the model biases. We found the different collections have minimal impact on the magnitude of the model biases (not shown). For the US2.2_ConfigA, the difference in the daytime biases is 0.9 K, and the difference is smaller for subsequent years; 0.4 K in 2014 (US2.2_ConfigB) and 0.1 K from 2015 (US2.2_ConfigC). It was

also found that the grassland sites, particularly Santa Rita Grassland, have a larger difference between the two collections (1.5 K smaller for the C6 retrieval) than the shrubland sites. The difference between the two collections was also found to be of a similar magnitude for the night-time retrievals, and is smaller than the overall variability in the night-time bias. It is important to recognise that the impacts of the retrieval algorithm are minimal when compared with the magnitude of the model biases being considered in this study.

Variability of surface temperatures could arise due to variability in cloud cover or soil moisture. In this study we consider only clear sky situations; both the model and observational datasets have been screened to remove cloud contamination, which suggests that soil moisture variability between the analysis years could be a factor for investigation. Point scale measurements of volumetric soil moisture at the eddy-covariance sites are made at depths of 5 cm and 15 cm. A six year multi-year mean soil moisture for each site and at each soil depth has been calculated, and used to calculate a soil

moisture anomaly. At both sites, the volumetric soil moisture in May is less than 0.05 kg m$^{-2}$ (0.10 kg m$^{-2}$) at 5 cm (15 cm) for all years in the evaluation. The *in situ* volumetric soil moisture measurements suggest that the moisture levels were almost always exhausted for each May analysis period and therefore it is unlikely there was sufficient soil moisture to impact on surface temperature variability.

In support of the eddy-covariance measurements, monthly 0.5°×0.5° soil moisture and soil moisture anomaly product from Climate Prediction Center (Fan et al. 2004) were used to assess the larger scale trends in soil moisture in southeastern Arizona. The soil moisture anomaly product indicates that May 2013 and 2014 were anomalously dry (-20 to -40 mm) for an extensive region of the western US, May 2015 had a neutral soil moisture anomaly, May 2016 and 2017 had localised dry regions confined within Arizona, and May 2018 was anomalously dry (-80 mm) for an extensive region of the western US.

**3.3 Correlation of LST biases with model orography and surface heterogeneity**

The LST biases were initially evaluated for the model diurnal cycle, and then extended to attribute observed trends in LST biases to changes to model parameters for a range of UM model configurations at four eddy-covariance flux sites. The discussion going forward will centre on a domain of 31.25-32.25 °N and 69.0-71.5 °W in southeastern Arizona in order to understand the spatial distribution of the surface temperature biases, and the mechanisms which give rise to the spatial distributions. The domain includes the San Pedro basin, Sulfur Springs Valley and San Simon Valley, consisting of shrublands, grasslands and riparian surfaces, as well as isolated, forested mountain ranges. The domain is heterogeneous in terms of surface cover and orographic slope and aspect with many model gridboxes and MODIS pixels including both craggy and forested or shrub land terrain within them.

Figure 4a shows the US2.2_ConfigA-E orography for the study domain. The figure demonstrates the complex terrain in the region with low-lying ground in the north west of the domain, numereous areas of mountianous terrain including both to the east and west of the Kendall Grassland and Lucky Hills with the highest mountain range of Chirichua range to the east (31.8 °N, 70.7 °W). The solar radiation reaching the surface is not considered to be uniform, and the absorption of solar radiation is highly dependent on local orography such as the orography slope and aspect (Manners et al. 2012). Figure 4b (c) presents the x-component (y-component) of the orographic slope which shows that the orography in this region is generally aligned in a north-south direction.

Firstly, we will investigate the correlation between the LST biases with the orographic slope. Our hypothesis being that northern and western-facing slopes of mountain ranges would have a shorter or delayed diurnal cycle due to reduced shortwave absorption at the surface and this could contribute to the spatial distribution of the LST bias. A linear least-squares regression is performed between the LST biases and the modelled orography (and surface fractional cover) and apply a Pearson product-moment correlation coefficient to measure the strength and direction of the linear relationship between two variables.

Figure 5a shows the spatial distribution of daytime LST biases between MODIS Terra (1755Z) collection 6 and the US2.2_ConfigA on 13 May 2013. This example was chosen to highlight typical LST biases seen during the daytime in cloud-free conditions. The mean LST bias with respect to MODIS collection 6 (collection 5) is -7.9±3.9 K (-7.8±3.7 K). The figure highlights the advantage of using the 1 km resolution LST from MODIS compared with the IASI 1D-Var retrievals presented in Figure 1 to examine the biases. There is significant variability in the distribution of bias with localised regions of warm and cold LST bias which would not be evident using a coarser retrieval. Secondly, we will examine correlations between the surface heterogeneity in terms of the US2.2 surface fractional cover and the spatial distributions of the LST biases.

Figure 5b presents the spatial distribution of the combined IGBP total grass fraction and IGBP shrub fraction in the study domain; and Figure 5c shows the spatial distribution of the IGBP bare soil cover fraction. We will investigate the coefficients of correlation between the LST biases and the vegetation and bare soil cover fractions represented in the US2.2_ConfigA-D surface fractional cover ancillary dataset.

Figure 5d-f presents the equivalent for the new ESA LC_CCI surface fractional cover introduced into the US2.2_ConfigE in 2018. Figure 5d shows the spatial distribution of daytime LST biases between MODIS Terra (1825Z) collection 6 and the US2.2_ConfigE on 30 May 2018. The mean LST bias with respect to MODIS collection 6 is -7.6$\pm$3.3 K. The mean LST bias for the domain is not significantly different to that seen in Figure 5a, although the spatial pattern is different, with localised cold and warm LST bias regions in different locations. This is predominantly due to a redistribution of the surface fractional cover in IGBP and the ESA LC_CCI datasets. Figure 5e presents the ESA LC_CCI spatial distribution the total grass fractions and shrub fractions; and Figure 5f shows the ESA LC_CCI spatial distribution of the bare soil fraction. The ESA LC_CCI reduces the total grass fractional cover and the bare soil fractional cover, and increases the shrub fraction across the domain. This results in closed shrub vegetation class. The ESA LC_CCI degrades the representation of the semi-arid ecosystem, in particular the representation of the bare soil cover fraction, which is reduced to 15-20 %, and is significantly below the observed fractions for this region (Scott et al., 2015).

The area average May mean surface temperature bias for the US2.2 for the study region for the six year analysis period has been calculated and presented in Figure 6a. For all configurations the night-time bias was less than 2.8 K, and suggests an improvement in the night-time bias between 2013 and 2018. The daytime biases are largest for 2013 and are progressively reduced between 2013 (US2.2_ConfigA) and 2015 (US2.2_ConfigC) from -8.2$\pm$4.4 K to -5.9$\pm$4.2 K (with respect to MODIS collection 6). In 2016 (US2.2_ConfigD), 2017 (US2.2_ConfigD) and 2018 (US2.2_ConfigE) the bias in the model increases to -7.2$\pm$4.7 K, -7.4$\pm$4.3 K and -7.6$\pm$3.3 K, respectively. This follows the same trend seen at the four eddy-covariance flux sites.

All data presented in Figure 6a has been cloud-screened in the US2.2 and for the MODIS overpasses. The impact of model cloud-clearing of the US2.2 has been assessed based on a threshold of total cloud cover greater than 0.1 (not shown); model cloud-clearing increases night-time biases in the order of 0.2-0.4 K, and reduces the absolute daytime biases between 0.3-0.5 K. Figure 6a presents the domain average LST bias using both MODIS C5 and C6 retrievals. There is a marginal colder bias with the MODIS collection 6 in the order of 0.5-0.6 K in 2013-2014 and less than 0.1 K in 2015 and 2016. The impact of the two MODIS collections on the correlation coefficients is minimal and only collection 6 is presented.

The coefficients of correlation between the LST bias and x-component (y-component) of the orographic slope have been calculated for the six year analysis period and are presented in Figure 6b (c). The solar illumination geometry of orography changes as a function of time of day, whilst the remotely sensed LST is a directional variable with each satellite platform (Terra and Aqua) maintaining the same angle with respect to the sun. Each platform measures a similar illumination geometry on each overpass, and therefore the coefficients of correlations are calculated separately for the Terra and Aqua retrievals in Figure 6b and 6c. The night-time coefficients of correlation have a value of +/-0.2 which indicates there is a relationship between the two variables, but it is weak and likely insignificant. For the x-component prior to 2018, the daytime coefficient

of correlation was positively correlated with a value of 0.41±0.05 (0.28±0.05) for Aqua (Terra) retrievals; and identifies that regions of cold model LST bias are found on easterly slopes and regions of warm model LST bias are found on westerly slopes. We find a stronger correlation between the x-component of the orographic slope and the LST bias for Aqua compared with Terra, whilst the difference between the two platforms was minimal for the y-component of the orographic slope.

The coefficients of correlation for the y-component of the orographic slope have weaker correlations of less than ±0.2 indicating there is no north-south difference in the bias, which may be because the orography in this region is generally aligned in a north-south direction.

          Our analysis also finds the coefficients of correlation relative to the y-component of the orographic slope at the time of the Terra overpasses are larger than at the time of the Aqua overpasses (not shown). This is an expected outcome of
the analysis as the Terra overpass is before noon and northern slopes will be cooler. No significant differences were observed for x-component of the orographic slope between the respective Terra or Aqua overpasses.

          The daytime and night-time correlation coefficients presented in Figure 6d indicate that from 2013–2017 there is a null to weak relationship (r less than -0.2) between the LST bias and distribution of the dominant grass vegetation. In 2018 (US2.2_ConfigE), with the introduction of ESA LC_CCI surface fractional cover, the coefficient of correlation becomes more
significant (0.33±0.07).

          More interesting are the correlation coefficients between the LST bias and bare soil cover fraction presented in Figure 6e. The LST bias is seen to have a moderate correlation in 2013-2017, with the IGBP bare soil cover fraction, during the daytime. The largest correlation is for 2013 with a correlation coefficient of 0.61±0.08 and this is also associated with the largest mean surface temperature bias of the six-year analysis. From 2014-2017, in configurations also using the IGBP surface
fractional cover, the correlation coefficients remain statistically significant with a range of 0.49±0.05 to 0.57±0.13. The correlation coefficients for the daytime overpasses suggest a moderately strong relationship that regions with a cold LST bias are associated with low bare soil cover fractions. At night the LST bias is weakly correlated (-0.21±0.06 in 2013) to the bare soil cover fraction with the correlation not significantly different from zero for 2014-2018.

          The 2013-2017 coefficient of correlations with the IGBP shrub surface fractions follow the same trend as the IGBP
bare soil fractions, although with a less significant correlation (0.36±0.09). Again, this suggests that the regions of cold model LST bias are additionally associated with regions with low shrub fractions (<20%), although it is secondary to the sensitivity of the bare soil. The results indicate that sparse vegetation canopies across the study domain are not well represented by the IGBP surface fractional cover. Our findings suggest that the development of surface cover ancillary datasets for sparse canopies is necessary.

In 2018, the coefficients of correlation are weaker for both the ESA LC_CCI shrub surface fractional cover (-0.31±0.04) and bare soil surface fractional cover (-0.30±0.06) and are the opposite sign to that calculated for the IGBP surface fractional cover. As described previously, the ESA LC_CCI degrades the representation of the semi-arid ecosystem, in particular the representation of the bare soil cover fraction, in forming a closed shrub vegetation class to represent the region. The ESA

LC_CCI bare soil fractions remain too low across the study domain, and is a possible explanation why the mean LST bias from the 2.2 km model for 2018 is $-7.6\pm3.3$ K and is effectively unchanged from the mean LST bias for 2013 ($-7.8\pm4.7$ K).

The coefficients of correlation with respect to the surface heterogeneity at the time of the Terra overpasses are larger than at the time of the Aqua overpasses (not shown). This is despite the LST biases not reaching a maximum until closer to the time of the Aqua overpass. This indicates that when the magnitude of the LST bias is at its maximum there is a competing cause to the LST bias which cannot only be fully explained by the representation of surface heterogeneity in the model.

## 3.4 Evaluation of UM surface energy balance

The total available energy at the surface is partitioned between the turbulent heat and moisture fluxes to the atmosphere, as well the ground heat flux to the soil, all of which ultimately control the surface temperature. Eddy-covariance measurements offer model verification of these surface exchange processes, and provide an opportunity to examine sources of model error by investigating components of the SEB simulated by the UM. Figure 7 presents scatter plots of observed SEB components compared with the US2.2_ConfigA SEB components for May 2013 at Kendall Grassland.

Figure 7a presents the net radiation (NR) for all sky conditions which represents the available energy at the surface from radiation; when NR is positive there is greater incoming radiation than outgoing radiation. At night, the NR term is negative as the net longwave is dominated by the outgoing terrestrial longwave flux. A night-time overestimate in NR of 36 W m$^{-2}$ is evident in the US2.2_ConfigA. The downwelling longwave (LWD) is also underestimated (not shown) which suggests that the night-time NR bias is caused by too much upwelling longwave, and potentially indicates the surface emissivity is too large. This result provides motivation for a revised bare soil emissivity. Daytime biases in NR are seen to be minimal at Kendall Grassland. Daytime biases are more significant at Lucky Hills (not shown) with an underestimation in the order of 16-25 W m$^{-2}$ which arises due to an underestimation of the downwelling shortwave radiation in the US2.2.

Turbulent transfer of heat and moisture towards (negative flux) or away (positive flux) from the surface within the atmosphere is represented by the sensible and latent heat fluxes, respectively. Figure 7b presents a scatter plot of the observed (corrected) sensible heat flux compared with the modelled sensible heat flux, which shows a positive model bias in the sensible heat flux of 25 W m$^{-2}$, and indicates the model flux is overestimated during the local solar maximum. During the transition period from early morning into the late morning period there is an underestimate in the modelled sensible heat flux, which suggests the US2.2_ConfigA does not represent the rate of increase in the sensible heat flux seen in the observations.

The equivalent scatter plot for the latent heat flux, presented in Figure 7c, indicates the US2.2_ConfigA latent heat fluxes are too large. A night-time (and transition) bias of 6 W m$^{-2}$, and a daytime bias of 23 W m$^{-2}$ were calculated. This result was also seen for GA/L3.1, as well as for the Lucky Hills site (plots not shown). In general it was found that there is a greater overestimate in the modelled turbulent heat and moisture fluxes when compared with the measured fluxes rather than the corrected turbulent fluxes.

Finally, Figure 7d presents the measured ground heat flux compared with the modelled ground heat flux. The night-time ground heat flux is well represented by the US2.2_ConfigA, however the transition and daytime ground heat flux is poorly

simulated by the US2.2_ConfigA. Again, this result was also seen for GA/L3.1. The US2.2_ConfigA daytime maxima is underestimated by 100 W m$^{-2}$ compared with the observations, however during the transition to morning and evening periods the ground heat flux is overestimated.

A delay of the onset of heating in the morning transition is evident in the observations which leads to a phase separation between the measured ground heat fluxes and the residual (NR–H–LE) (plot not shown), which possibly casts doubt on the ground heat flux measurements at Kendall Grassland. The timing of the measured ground heat flux is poorly represented, relative to the turbulent and radiant forcing, which suggests to a possibility that the measurements, taken at depth, have not been correctly extrapolated to the surface. However, an alternative interpretation could be that at the Kendall Grassland site there is shading at location of the ground heat flux plates from vegetation, whilst the net radiometers are mounted above the vegetation canopy and not subject to the effects of shading, which could lead to the lag in the ground heat flux relative to the radiative forcing.

This helps explain the timing hysteresis observed in Figure 7b in the corrected sensible heat flux, which is seen to be in the opposite direction to the ground heat flux; closure of the surface energy balance has been forced and therefore any lag in the timing of the measured ground heat flux will propagate into the corrected sensible heat flux.

Our results indicate the models' fluxes at Kendall Grassland (and Lucky Hills) are deficient in representing ground heat fluxes, which suggests that the excess modelled turbulent heat and moisture fluxes are compensated for with an underestimate in the modelled ground heat flux. This result indicates the partitioning of the turbulent heat and moisture fluxes to the atmosphere, and the flux of heat to the soil are not well represented in the US2.2 (and GA/L3.1), and could contribute to the surface temperature biases evaluated in this study. A comprehensive evaluation of the surface energy balance of the Unified Model and the standalone JULES land surface model in necessary to understand the model errors in greater detail, although this is out of scope for this study.

## 4 Conclusions

A limitation of the Met Office operational data assimilation scheme is that surface-sensitive infrared hyperspectral satellite sounding channels cannot be used during daytime periods where biases in the Numerical Weather Prediction (NWP) model background land surface temperature (LST) are greater than 2 K. The Met Office Unified Model (UM) has a significant cold bias in LST in semi-arid regions when compared with satellite observations. This work evaluates UM surface temperature biases for two UM global configurations, Global Atmosphere/Land 3.1 (GA/L3.1) and Global Atmosphere/Land 6.1 (GA/L6.1) and in a Limited Area Model (LAM) at 2.2 km (US2.2) resolution for a study domain in southeastern Arizona USA which coincided with the SALSTICE (Semi-Arid Land Surface Temperature and IASI Calibration Experiment) campaign

The UM surface temperature biases for the North American continent during May, the time of maximum LST biases, were investigated with IASI 1D-VAR retrievals. GA/L3.1 gave rise to an east-west divide in the magnitude of LST biases with cold biases in excess of -10 K in the south-west US, western Mexico and extended east into the Great Plains. Moderate LST

biases, in the range of -4 to -6 K, were shown to extend into the northern US. The LST bias was found to be reduced in GA/L6.1 compared with GA/L3.1, although regional biases such as the south-west US were still prominent.

The UM surface temperature biases were examined at higher resolution using MODIS surface temperature retrievals from the Aqua and Terra platforms for an analysis period of May 2013-May 2018. The evaluation was in conjunction with ground-based measurements from eddy-covariance flux tower sites in the Walnut Gulch Experimental Watershed and Santa Rita Experimental Range in southeastern Arizona. Examining the representation of the diurnal cycle of surface temperature, it was found that in GA/L3.1 biases in modelled LST were largest in the mid-morning, which indicated GA/L3.1 struggled to capture the magnitude of the warming from the morning transition to the late morning period. The phase of the diurnal cycle of surface temperature in GA/L6.1 showed a significant improvement relative to GA/L3.1, and supported the result found relative to IASI 1D-VAR retrievals for the North American continent. The diurnal cycle in the higher resolution US2.2, also showed that the phase of the surface temperature was improved relative to the GA/L3.1 configuration and improved the timing of the initial warming during the morning transition.

The surface temperature bias response for different vegetation biomes was investigated at four eddy-covariance flux tower sites located in different land classification types. The improvement in surface temperature in the US2.2 (compared with the global configuration) was found to be greater at the two shrubland sites, Lucky Hills and Santa Rita Mesquite, compared with the two grassland sites, Kendall Grassland and Santa Rita Mesquite. Improvements at all four sites in the US2.2 was attributed to changes in the bare soil parameters including a revised bare soil emissivity and revised thermal and momentum roughness lengths for bare soil. The shrubland sites had an increase in the bare soil fractional cover associated with the increasing model resolution, and increased the sparsity of the vegetation cover, and hence improved the model representation of the surface heterogeneity. In contrast, at the grassland sites, there was a reduction in bare soil fractional cover.

The limitation of available water for vegetation in semi-arid regions results in a very heterogeneous natural landscape, which increases the scientific challenges of representing such surface heterogeneities in land surface models. Our study examined a domain in southeastern Arizona in order to understand the spatial distribution of the surface temperature biases, and the mechanisms which give rise to the spatial distributions. The study domain is heterogeneous, in terms of surface vegetation cover and orographic slope and aspect, with many model gridboxes including both craggy and forested or shrub land terrain within them. Our results highlight there was no dominant underlying cause to the distribution of LST biases in the study domain. The LST bias was found to have a moderate correlation with the International Geosphere-Biosphere Programme's (IGBP) bare soil cover fraction during the daytime and suggested that regions of cold model LST bias were associated with low bare soil cover fractions. Coefficients of correlation with the IGBP shrub surface fractions were found to follow the same trend as the IGBP bare soil fractions, although with a less significant correlation, and secondary to the sensitivity of the bare soil. The results indicate that sparse vegetation canopies are not well represented by the IGBP surface fractional cover.

Considering orography in the study domain, the daytime coefficients of correlation were positively correlated with the x-component of the orographic slope, which indicated that regions of cold model LST bias were found on easterly slopes and

regions of warm model LST bias were found on westerly slopes. The coefficients of correlation for the y-component of the orographic slope were found to have weaker correlation of less than ±0.2 as the orography in the study region is generally aligned in a north-south direction.

For the US2.2 in 2018, the surface fractional cover ancillary used the European Space Agency's Land Cover Climate Change Initiative (ESA LC_CCI) global vegetation distribution mapped to the JULES five PFTs. The ESA LC_CCI ancillary degrades the representation of the semi-arid ecosystem in the study region, in particular the representation of the bare soil cover fraction, which was reduced to 15-20 %, and is significantly below the observed fractions for this region (Scott et al., 2015). The ESA LC_CCI bare soil fractions were shown to be too low across the study domain, even more so than the IGBP bare soil cover fractions, and is a possible explanation why the mean LST bias in the US2.2 for 2018 was -7.6±3.3 K and effectively unchanged from the mean LST bias for 2013 (-7.8±4.7 K).

The US2.2 was found to be deficient in representing ground heat fluxes when compared against eddy-covariance measurements at Kendall Grassland and Lucky Hills sites. The modelled turbulent heat and moisture fluxes were overestimated compared with observations. The modelled latent heat flux was overestimated for all periods of the diurnal cycle, and the modelled sensible heat flux was overestimated during the local solar maximum. This result indicates the partitioning of the turbulent heat and moisture fluxes to the atmosphere, and the flux of heat to the soil are not well represented in the US2.2 (and GA/L3.1), and could contribute to the surface temperature biases evaluated in this study. Our results call for a comprehensive evaluation of the SEB of the Unified Model and the standalone JULES land surface model in semi-arid regions.

The validation presented in this paper used ground-based and satellite measurements and to a large degree, the two comparisons generate comparable results considering the vast differences in scales of the measurements. The two methods have advantages and disadvantages that complement each other; the MODIS comparisons gave a high spatial resolution representation at specific snapshots in time while the eddy-covariance site measurements gave full diurnal cycles although with very limited areal coverage. The MODIS data are conducive to geostatistical analysis while the ground site data is suitable for time-series analysis. A further consideration is the disparity between the footprint size of the IRTs, radiation measurements and the ground heat flux plates relative to those of the sonic anemometers measuring the turbulent fluxes.

With recent advances in supercomputing power, the ability to perform high resolution ensemble forecasting, for example within a research LAM such as the US2.2, is becoming viable. This will provide an opportunity to evaluate the impact of forecast uncertainty on the land surface processes, rather than only for the deterministic forecast as has been carried out in this study. The Met Office Global and Regional Ensemble Prediction System (MOGREPS) is the ensemble system that produces uncertainty information for the model configurations.

The outcomes of SALSTICE show the difficulties in producing land-surface temperatures that match the current state-of-the-art satellite retrievals within our current NWP system. The unfortunate fact is that LST is not used to evaluate the model during model development and much of the LST information available from satellite is thrown away by the data assimilation system. It is not surprising therefore that the prediction of LST in our operational NWP suite has not improved significantly since GA/L3.1.

**Code availability**

*Obtaining the UM.*

The Met Office Unified Model is available for use under licence. A number of research organisations and national meteorological services use the UM in collaboration with the Met Office to undertake basic atmospheric process research, produce forecasts, develop the UM code and build and evaluate Earth system models. For further information on how to apply for a licence see http://www.metoffice.gov.uk/research/collaboration/um-collaboration.

*Obtaining JULES.*

JULES is available under licence free of charge. For further information on how to gain permission to use JULES for research purposes see https://jules.jchmr.org/software-and-documentation.

**Data availability**

Data used in this paper are available at the Ameriflux Data Repository (http://ameriflux.lbl.gov/).

**Author contribution**

JKB evaluated the model simulations and wrote the manuscript. RCH was the project investigator for SALSTICE. RLS supplied eddy-covariance data. MJB and JME provided advice on the surface energy balance of the UM. MW developed the 2.2 km model configuration. JCT advised on the radiative transfer calculations for calculating the downwelling longwave applied to the IRT dataset. All authors were involved in discussions throughout development, and all authors commented on the paper.

**Competing interests.**

The authors declare that they have no conflict of interest.

**Acknowledgements**

This work was funded by the Met Office. The author would like to thank the efforts of staff at USDA-Agricultural Research Service's Southwest Watershed Research Centre for providing the eddy-covariance data and facilitating the placement of additional sensors at Lucky Hills and Kendall Grassland within the Walnut Gulch Experimental Watershed.

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

| | | Global | US2.2 |
|---|---|---|---|
| | x | GA/L3.1 | US2.2_ConfigA |
| 2013 | i) Dynamics | NewDynamics (Walters et al., 2011) | NewDynamics (Walters et al., 2011) |
| | ii) Resolution | 25 km horizontal resolution | 2.2 km horizontal resolution |
| | iii) DA bias correction | Static bias correction | No data assimilation |
| | iv) Initialisation | - | US4.4, T+3 |
| | v) Land cover | IGBP land cover | IGBP land cover |
| | vi) Bare soil parameters | ε=0.97, $z_{OM}$=0.0032 m, $z_{OH}/z_{OM}$=0.10 | ε=0.97, $z_{OM}$=0.001 m, $z_{OH}/z_{OM}$=0.02 |
| | x | GA/L3.1 | US2.2_ConfigB |
| 2014 | i) Dynamics | NewDynamics (Walters et al., 2011) | NewDynamics (Walters et al., 2011) |
| | ii) Resolution | 25 km horizontal resolution | 2.2 km horizontal resolution |
| | iii) DA bias correction | Static bias correction | No data assimilation |
| | iv) Initialisation | - | US4.4, T+3 |
| | v) Land cover | IGBP land cover | IGBP land cover |
| | vi) Bare soil parameters | ε=0.97, $z_{OM}$=0.0032 m, $z_{OH}/z_{OM}$=0.10 | ε=0.90, $z_{OM}$=0.001 m, $z_{OH}/z_{OM}$=0.02 |
| | x | GA/L6.1_17km_static | US2.2_ConfigC |
| 2015 | i) Dynamics | ENDGame (Walters et al., 2016) | ENDGame (Walters et al., 2016) |
| | ii) Resolution | 17 km horizontal resolution | 2.2 km horizontal resolution |
| | iii) DA bias correction | Static bias correction | No data assimilation |
| | iv) Initialisation | - | GA/L6.1_17km_static, T+0 |
| | v) Land cover | IGBP land cover | IGBP land cover |
| | vi) Bare soil parameters | ε=0.90, $z_{OM}$=0.001 m, $z_{OH}/z_{OM}$=0.02 | ε=0.90, $z_{OM}$=0.001 m, $z_{OH}/z_{OM}$=0.02 |
| | x | GA/L6.1_17km_VarBC | US2.2_ConfigD |
| 2016 | i) Dynamics | ENDGame (Walters et al., 2016) | ENDGame (Walters et al., 2016) |
| | ii) Resolution | 17 km horizontal resolution | 2.2 km horizontal resolution |
| | iii) DA bias correction | Variational bias correction (VarBC) | No data assimilation |
| | iv) Initialisation | - | GA/L6.1_17km_VarBC, T+0 |
| | v) Land cover | IGBP land cover | IGBP land cover |
| | vi) Bare soil parameters | ε=0.90, $z_{OM}$=0.001 m, $z_{OH}/z_{OM}$=0.02 | ε=0.90, $z_{OM}$=0.001 m, $z_{OH}/z_{OM}$=0.02 |
| | x | GA/L6.1_17km_VarBC | US2.2_ConfigD |
| 2017 | i) Dynamics | ENDGame (Walters et al., 2016) | ENDGame (Walters et al., 2016) |
| | ii) Resolution | 17 km horizontal resolution | 2.2 km horizontal resolution |
| | iii) DA bias correction | Variational bias correction (VarBC) | No data assimilation |
| | iv) Initialisation | - | GA/L6.1_17km_VarBC, T+0 |
| | v) Land cover | IGBP land cover | IGBP land cover |
| | vi) Bare soil parameters | ε=0.90,$z_{OM}$=0.001 m, $z_{OH}/z_{OM}$=0.02 | ε=0.90, $z_{OM}$=0.001 m, $z_{OH}/z_{OM}$=0.02 |
| | x | GA/L6.1_10km_VarBC | US2.2_ConfigE |
| 2018 | i) Dynamics | ENDGame (Walters et al., 2016) | ENDGame (Walters et al., 2016) |
| | ii) Resolution | 10 km horizontal resolution | 2.2 km horizontal resolution |
| | iii) DA bias correction | Variational bias correction (VarBC) | No data assimilation |
| | iv) Initialisation | - | GA/L6.1_10km_VarBC, T+0 |
| | v) Land cover | IGBP land cover | ESA Land Cover CCI |
| | vi) Bare soil parameters | ε=0.90, $z_{OM}$=0.001 m, $z_{OH}/z_{OM}$=0.02 | ε=0.90, $z_{OM}$=0.001 m, $z_{OH}/z_{OM}$=0.02 |

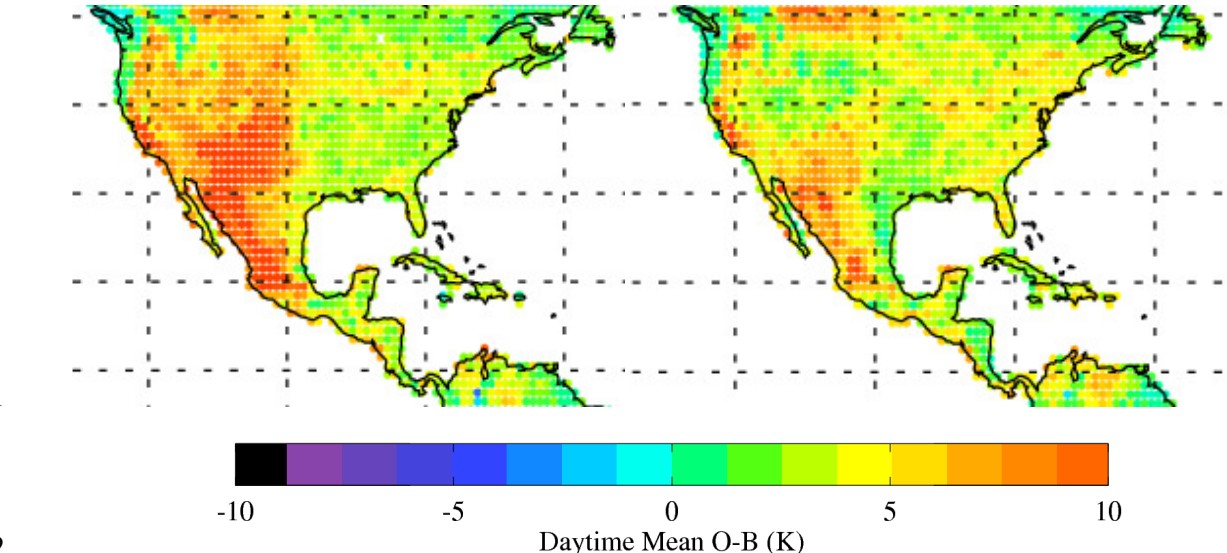

**Figure 1. UM surface temperature biases (observed-minus-background, O-B) compared to IASI 1D-VAR retrievals for the North American continent during (left) GA/L3.1, May 2013 and (right) GA/L6.1_17km_static, May 2015.**

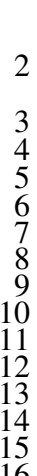
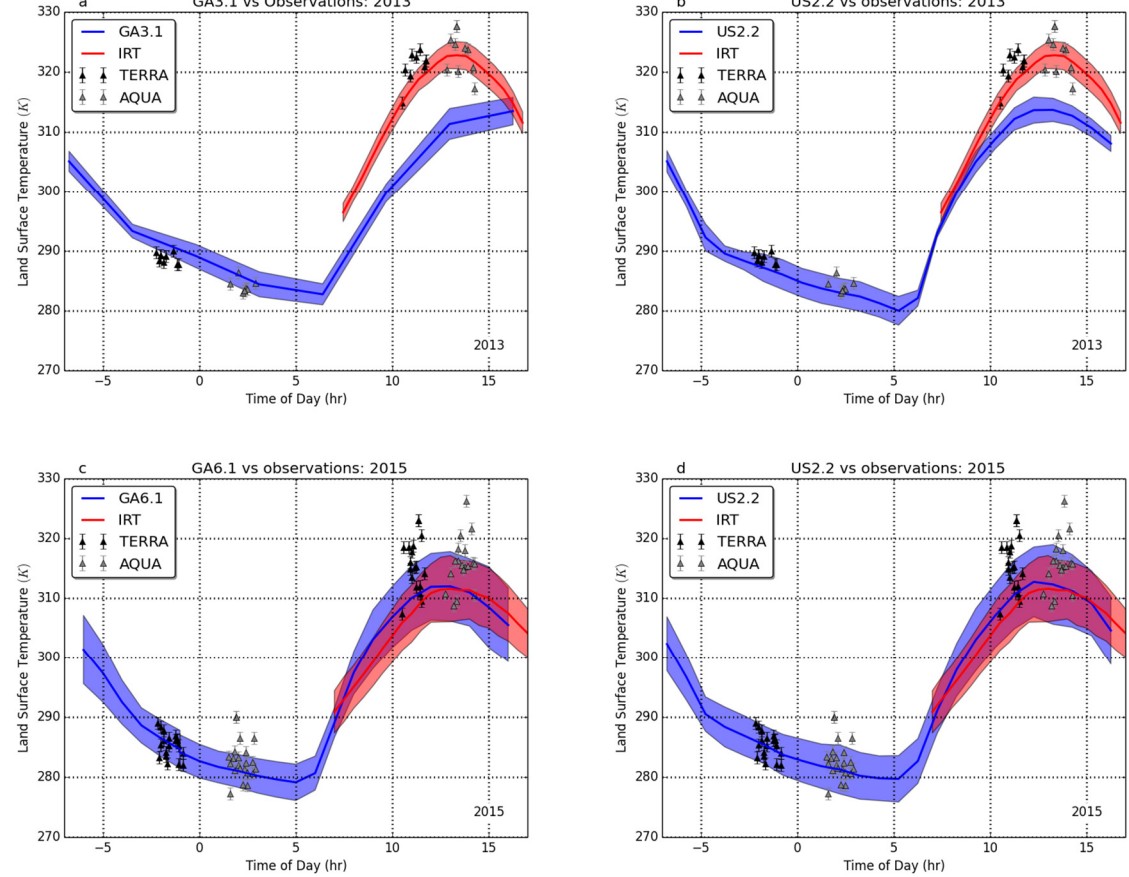

**Figure 2. Diurnal cycles of surface temperature at Kendall Grassland (*red*) observed from IRT measurements compared with (*blue*) UM configurations. The (*red shading*) is the standard deviation of the IRT measurements and (*blue shading*) is the standard deviation of the model data. (a) GA/L3.1; (b) US2.2_ConfigA; (c) GA/L6.1_17km_static; and (d) US2.2_ConfigC. Overlaid retrievals are (*black triangle*) TERRA LST (*grey triangle*) AQUA LST. Time is Local Standard Time. N.B The IRT measurements have only been plotted from 6 am to 6 pm.**

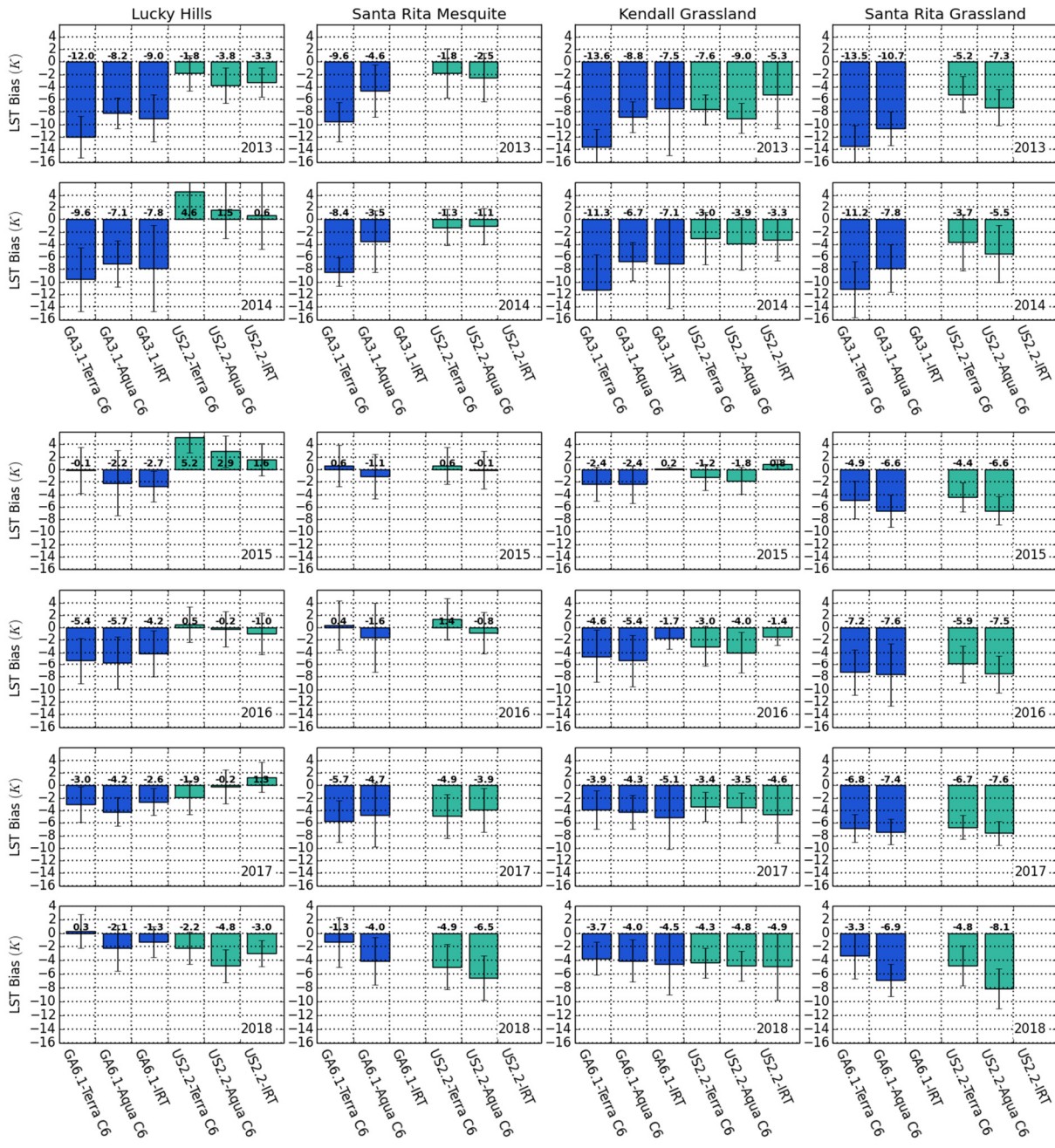

**Figure 3 Daytime LST biases from the online UM configurations compared with MODIS C6 Terra and Aqua LST**
**retrievals at the four eddy-covariance flux sites; Lucky Hills, Santa Rita Mesquite, Kendall Grassland and**
**SantaRita Grassland. Daytime LST biases from the online UM configurations compared with IRT observations**
**are presented for Lucky Hills and Kendall Grassland sites. In blue the GA/L3.1 configurations (2013, 2014) and**
**GA/L6.1 configurations (GA/L6.1_17km_static, 2015); (GA/L6.1_17km_VarBC, 2016 and 2017); and**
**(GA/L6.1_10km_VarBC, 2018). In cyan are the US2.2_ConfigA-E configurations. The LST evaluation has**
**performed for six years in (row 1) 2013; (row 2) 2014; (row 3) 2015; (row 4) 2016; (row 5) 2017; and (row 6) 2018.**

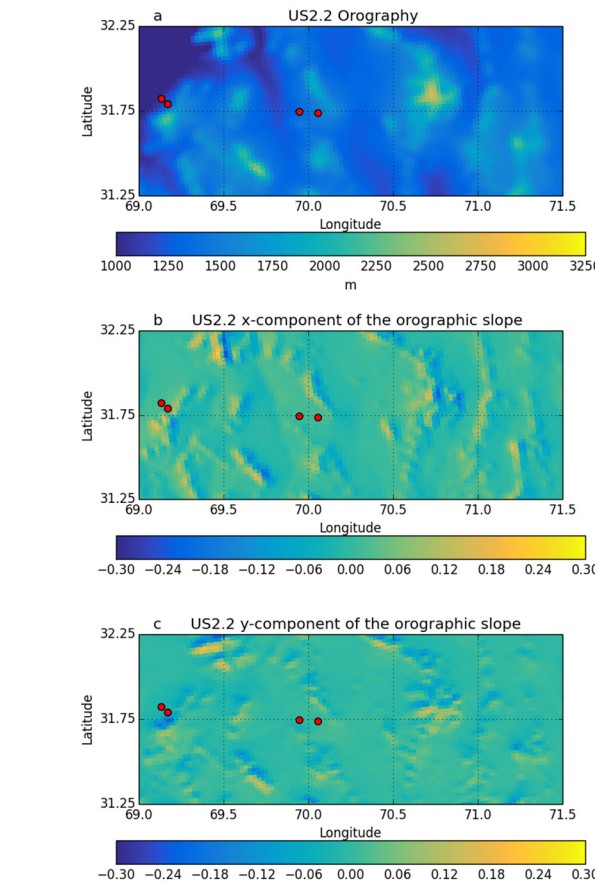

**Figure 4. (a)** Model orography for US2.2_ConfigA-E configurations. **(b)** US2.2 x-component of the orographic slope, where negative values indicate easterly facing slopes, and positive values indicate westerly facing slopes. **(c)** US2.2 y-component of the orographic slope, where negative values indicate northerly facing slopes, and positive values indicate southerly facing slopes. **(Red dots)** Lucky Hills (31.75 ºN, 110.05 ºW), Kendall Grassland (31.73 ºN, 109.94 ºW), Santa Rita Mesquite (31.82 ºN, 110.87 ºW), and Santa Rita Grassland (31.79 ºN, 110.83 ºW).

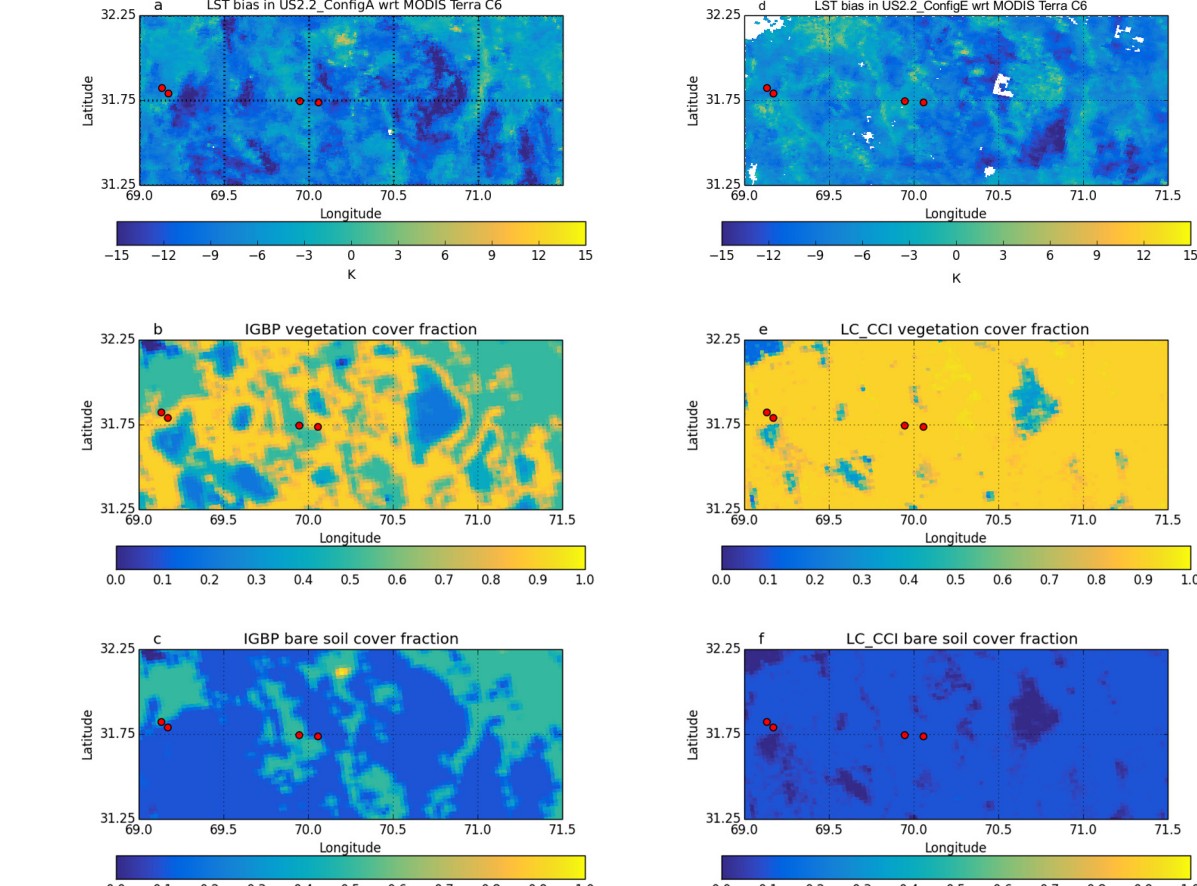

Figure 5. (a) Spatial distribution of surface temperature biases in the US2.2_ConfigA with respect to MODIS Terra Collection 6 on 13 May 2013. (b) IGBP total grass (C3 and C4) and shrub cover fraction for US2.2 configuration. (c) IGBP bare soil cover fraction for US2.2 configuration. (d) Spatial distribution of surface temperature biases in the US2.2_ConfigE with respect to MODIS Terra Collection 6 on 30 May 2018. (e) LC_CCI total grass (C3 and C4) and shrub cover fraction for US2.2 configuration. (f) Bare soil cover fraction for US2.2 configuration. (Red dots) Lucky Hills (31.75 ºN, 110.05 ºW), Kendall Grassland (31.73 ºN, 109.94 ºW), Santa Rita Mesquite (31.82 ºN, 110.87 ºW), and Santa Rita Grassland (31.79 ºN, 110.83 ºW).

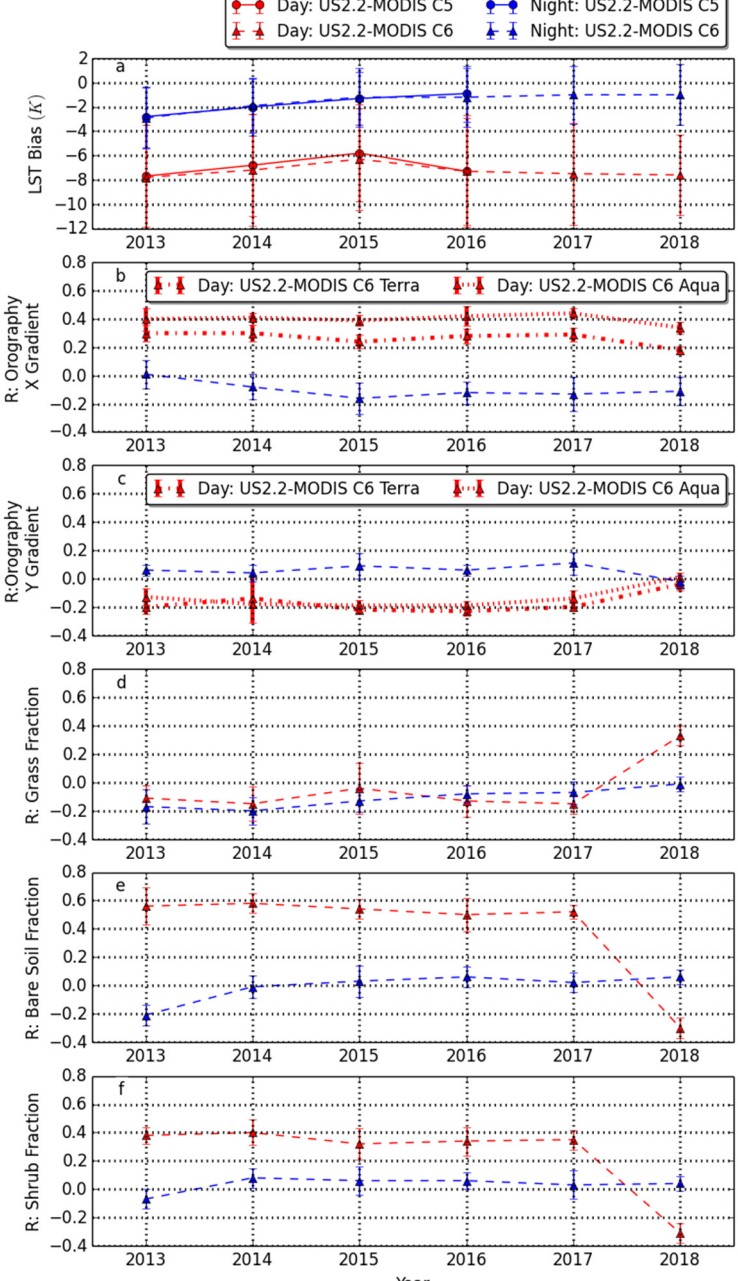

**Figure 6 (a) US2.2 (red) daytime and (blue) night-time LST biases determined from MODIS (solid) collection 5 and**
**(dashed) collection 6 overpasses (the average of Terra and Aqua retrievals) acquired during May 2013 – 2018. The**
**coefficient of correlation (r) for daytime and night-time during May 2013 – 2018 between the surface temperature bias**
**and (b) US2.2 x-component of the orographic slope; (c) US2.2 y-component of the orographic slope; (d) grass**
**fractional cover; (e) bare soil fractional cover; and (f) shrub fractional cover for US2.2 configuration. N.B In panel b (c)**
**the collection 6 (dotted) Terra and (dot-dashed) Aqua retrievals are separated for presenting the correlations with the x-**
**component (y-component) of the orographic slope.**

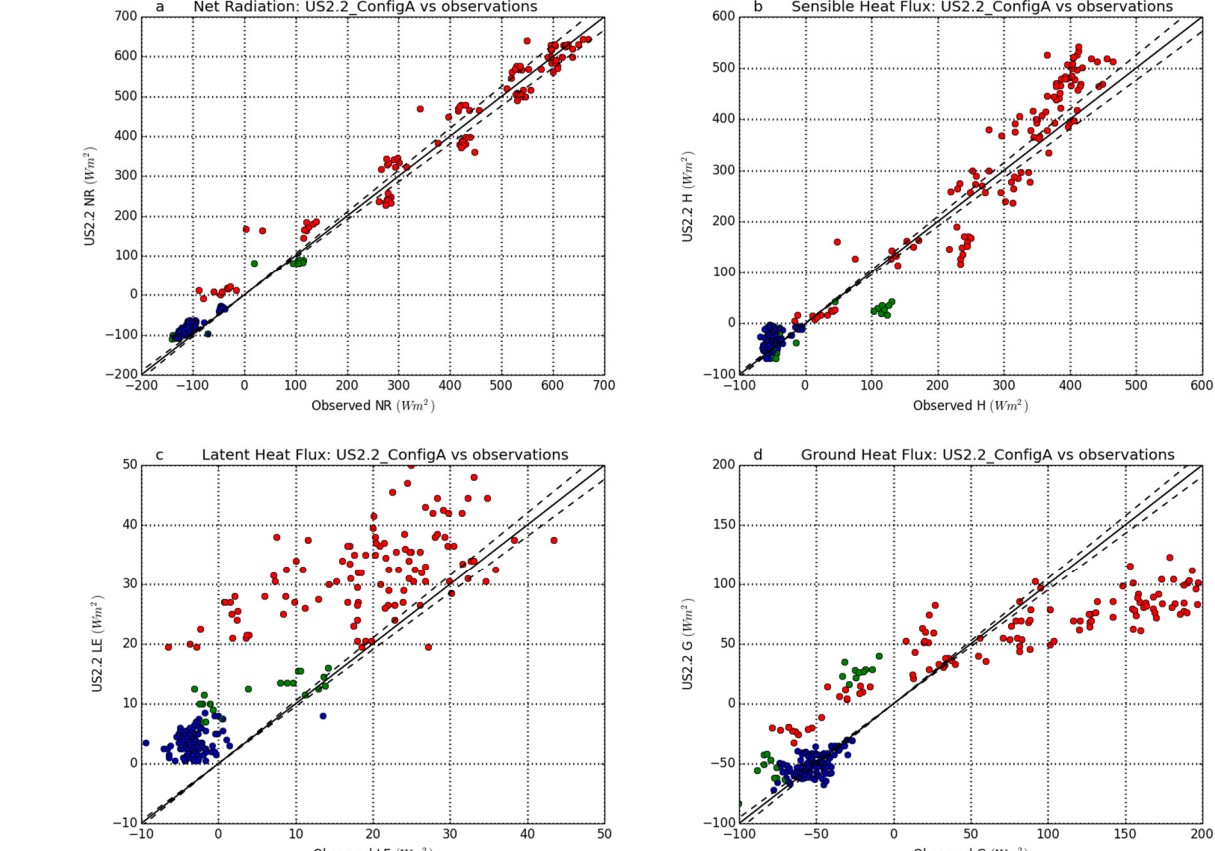

**Figure 7. Scatter plots comparing the observed components of the surface energy balance with the modelled US2.2_ConfigA components of the surface energy balance at Kendall Grassland for May 2013. The panels from top to bottom are (a) net radiation, (b) (corrected) sensible heat flux, (c) (corrected) latent heat flux, and (d) (measured) ground heat flux. The components of the SEB are separated into (blue) night-time (model SWD < 5 W m$^{-2}$), (green) transition (model SWD 5 – 200 W m$^{-2}$), and (red) daytime (model SWD > 200 W m$^{-2}$).**

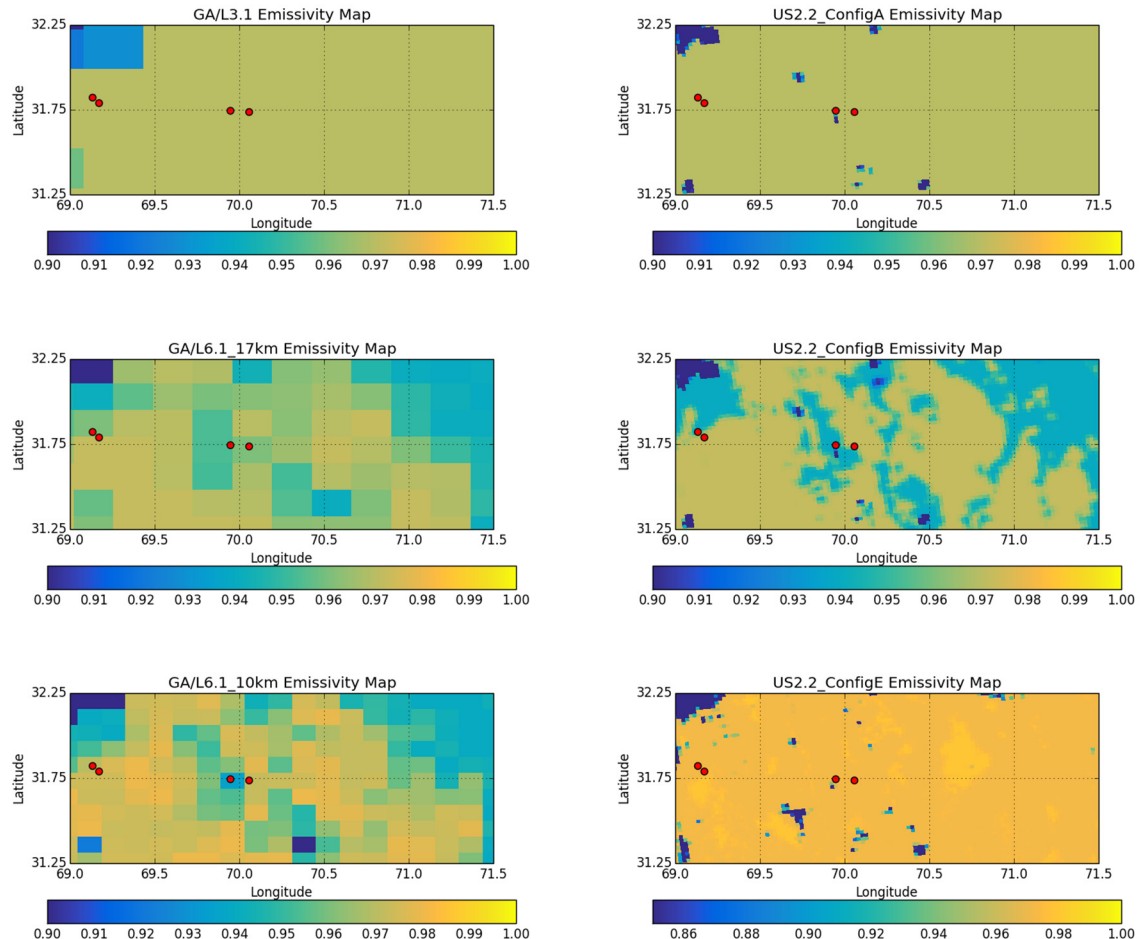

**Supplementary Figure 1. Emissivity map of the study area (a) GA/L3.1; (b) GA/L6.1_17km; (c) GA/L6.1_10km; (d) US2.2_ConfigA; (e) US2.2_ConfigB; and (f) US2.2_ConfigC.**