# Peer review of "Evaluating the Met Office Unified Model land surface temperature in Global Atmosphere/Land 3.1 (GA/L3.1), Global Atmosphere/Land 6.1 (GA/L6.1) and Limited Area 2.2km configurations."

_Geoscientific Model Development, 2018_

## Referee Comment (RC1) · Anonymous Referee #1 · 25 Jan 2019

The manuscript provides an assessment of land surface temperature (LST) simulated with different configurations of the UK Met Office Unified model. The exercise is made for a small area in the US (Arizona) taking advantage of simulations and data gathered for a particular experiment (SALSTICE), which was focused on model LST bias with respect to IASI retrievals. Model LST simulations are compared with IASI and MODIS products, as well as with in situ estimates. Model net radiation, turbulent heat fluxes at the surface and ground flux are also compared with ground observations.

[Figure]

The manuscript is very well written and the subject is of interest, given the limitation in the assimilation of radiances sensitive to lower troposphere over land due to the large model skin temperature biases. However, it is difficult to draw solid conclusions when different model configurations (in terms of dynamics, resolution, approach to bias correction, surface parameters) are not run for a common period. I suggest the article to be accepted subject to revisions in line with my comments below.

1) On local estimates of LST (section 2.2.2): I fully agree with the need to account for the uncertainty in local emissivity for the LST ground estimates. From the description provided in this section, it seems you do not correct the surface leaving radiance for the downward radiation that is reflected by the surface. This may be the same order of emissivity uncertainty for the 8-14 micro-m band. Please check and modify the data and model versus in situ comparisons in the manuscript as needed.

2) End of section 2.3 (page 6): The angular dependence of LST estimates should not be linked to atmospheric effects, as these should have been corrected during the retrieval process. Although insufficient correction for the optical path may still persist, the effects described in the text are more frequently a consequence of spatial heterogeneity (i.e., different viewing perspectives may actually yield different scenes, even if matching in time and space) and therefore are essentially dependent on the viewing & illumination geometry. That is why angular-dependent biases are mostly inexistent for night-time observations.

3) Lines 3-8 (page 8): Indicate how the change in the emissivity attributed to each tile (bare ground, grasses, . . .) change the emissivity map over the study area. I'd say that overall you have a slight decrease for GA/L6.1 and US2.2(A to D) and an increase in US2.2E due to the drastic reduction of bare ground fraction.

4) Lines 10-15 (page 8): Please include a short justification for the use of different ZoH/ZoM ratios in the global and limited area model versions.

5) On the overall analysis of model simulations: As referred above in my general com-

ment, the results of different model configurations correspond to the same period of the year (May), but for model runs performed for different years. You must ensure that when comparing these results, they are not affected by inter-annual variability. In order words, please show that the conditions observed in each May of the 2013-2018 period do not deviate greatly from the average. In the case they do, please check how that may have affected your results. This is relevant, since a dryer or rainier than usual year may lead to a significant change in vegetation cover (and therefore in surface parameters such as surface albedo and emissivity, and even ZoM), soil moisture availability (and likely in the partition between latent and sensible heat fluxes), which will certainly impact your model performance.

6) Last line of page 9 – line 7 of page 10: I'm not sure I follow what is meant here, especially with what respects the degradation in the representation of the grassland fractions. In contrast to the latter, when use a higher resolution landcover, you get better representation for bare soils: is this so? Why? Please clarify (or just rephrase).

7) When discussing the statistics between the various model configurations and MODIS LST products (collections 5 and 6), it would be useful to have an idea how both compare with the in situ estimates (please make sure these are properly estimated, as commented above). You may consider adding a table with a summary of all these, including an average of the in situ (or MODIS) LST per site, which would somehow answer my question above on stable the conditions are among the studied years. This may also help you check if there are years/sites for which MODIS (Aqua or Terra) presents higher biases, and therefore help you analysing your model comparisons with MODIS LST.

8) On the assessment of model biases and terrain slope: The impact of slope, especially the x-component) surely differs for Terra (morning overpass) and Aqua (afternoon overpass), if this is essentially related to the LST contrast between slopes facing/hiding from the sun. Maybe this effect is more noticeable in the afternoon, and in that case the "Aqua signature" prevails. In any case, the illumination geometry is obviously very

relevant for this, and therefore results should be assessed for the two platforms separately.

9) Lines 15-16 of page 14 "Our findings suggest that the daytime model LST bias could be minimised by increasing the bare soil cover fraction in the study regions". I don't think you can say this, as you are suggesting that you should change the fraction of bare soil, instead e.g., correcting, e.g., model parameters where the fraction of bare soil is low.

10) The comparisons between model and observed net radiation and surface energy fluxes are only discussed for a single site/year. Although the issue of different models run for different periods could make the discussion difficult, it would be interesting to know how the comparison between simulations and observations evolved as the model land surface temperature changed.

11) Editorial:

- Abstract: "The diurnal cycle of LST in Global Atmosphere/Land 6.1 (GA/L6.1) showed a significant improvement relative to GA/L3.1": Please be more specific (meaning quantitative) here.

- lines 5-6 (page 6): Suggest replacing "to give site-specific LST for each site." by "to give site-specific LST.".

- Figures 4 and 5: suggest the authors include a short title for each panel (e.g., LST bias – US2.2A), to facilitate their interpretation.

- line 23 (page 12): "IASI"

- Figure 7: Please ensure each individual scatter-plot has the same range in the y- and x-axes, since we are comparing the same variable (model versus observations). For the same reason, please resize the diagrams so that they are closer to a square, i.e., so that the length in the y-axis corresponding to, say, 10 wm-2 roughly matches the same length for 10 Wm-2 in the x-axis.

- Line 12 page 18: Please rephrase sentence.

---

## Referee Comment (RC2) · Anonymous Referee #2 · 28 Jan 2019

The manuscript "Evaluating the Met Office Unified Model Global Atmosphere/Land 3.1 (GA/L3.1) and Global Atmosphere/Land 6.1 (GA/L6.1) land surface temperature. Outcomes of the SALSTICE campaign" by Brooke et al. describes an investigation of land surface temperature biases using the Met Office Unified Model. Overall, the results are interesting and show aspects of the errors in simulated land surface temperature for a number of different model configurations, and would be of interest to the scientific community. The simulated temperature biases are related to a number of different

parameters in the model. ÂăFor the most part, the manuscript is well written, although some parts could provide more motivation and be made clearer to the reader. Most of my concerns are relatively minor and should be straight forward to address and the manuscript should be acceptable for publication in Geoscientific Model Development after these concerns are addressed.

Major comments:Âă 1. The title is a bit misleading, given that it only mentions two model configurations while limited area models are also applied in the study. In addition, the title mentions SALSTICE, but it is not clear to me if any SALSTICE data is included. Was the deployment of the eddy-covariance systems considered part of that study, or was SALSTIC only the airborne deployment that is not used at all. 2. The application of cloud screening applied in the study needs to be described better. It is mentioned in a couple of places, but I think that it has an important impact on the interpretation of the satellite derived LSTs and should be presented in a consistent way, perhaps in the Methodology section. Âă 3. A little more background is needed to help the reader understand the need for all of the model configurations that are presented. I believe that one reason is related to changes in model configurations with time, while other differences are related to parameter values.Âă 4. The authors should add a few notes regarding some of the calculations. For example, how is the bias computed? Is the correlation coefficient Person's correlation coefficient or something different?

Minor comments: 1. Page 1, line 12: Should "greater than 2 K", be "greater than 2 K in magnitude"? 2. Page 1, line 13: This is related to my major comment 3 and minor comment 13. A number of different model configurations are used in the study, and it is hard to see the reason why in the abstract. If there is space (given the word limit of the abstract) some reasons for application of different model configurations would be helpful. 3. Page 1, line 18. Please define "Terra" on first usage. 4. Page 2, line 7-9. The sentence describing the IASI was confusing as written. It seems to imply that the data is never assimilated, but other part of the manuscript seem to describe that these observations are not used only when the errors are large. 5. Page 2 lines 10-17.

Could this paragraph be adjusted for those that are not completely up-to-date with the UM and other models used by the Met Office? The second sentence says that LST is not assimilated into the UM, but the next sentence talks about LSTs being applied in the Met Office operational model.Âă 6. Page 2, line 18. What is meant by "background" in this context?Âă 7. Page 2, line 31-33. The surface albedo plays an important role in the surface energy budget. Should albedo also be mentioned in this paragraph?Âă 8. Page 4, line 4-6. Why are two different time periods used in this study? I recognize that it is, at least in part, due to the timing of th SALSTICE study. Are the eddy-covariance measurements only available for the shorter time? 9. Page 4, line 31-33. Could there also be errors associated with the representativeness of the soil heat flux? You mention this later, but it would also fit here. 10. Page 5, line 4. Is there any need to consider clouds in the IRT measurements in order to ensure that they are consistent with the satellite observations? I see something mentioned in section 2.3, but should it also be mentioned here?Âă 11. Page 5, line 18-22. Is there a reason why you would expect the nighttime values to be unreliable, but not the daytime values?Âă 12. Page 6, line 8-9. I can understand why you wanted to include information about the cloud clearing here, but would it make more sense in the modeling section?Âă 13. Page 6, line 19. Section 2.4 could be improved with additional background information regarding the selection of the various model configurations. Why are so many model configurations used? Why are both global and regional models used? I believe that one reason are new versions of the operational model.Âă Table 1 helps, but probably isn't sufficient.Âă 14. Page 8, line 19-21. Is the sign convention the same in Figure 1 and the text?Âă Based on the figure it looks like nearly all of the biases are positive, not negative.Âă 15. Page 9, line 11-14. I agree that the field of view of the IRTs is much smaller than the size of the model grid cell, yet the color shading still shows good agreement between the simulations and the IRT.Âă 16. Page 9, line 21-22. I agree that high resolution data sets are likely important, but could other factors also lead the improved performance at higher resolution? For example, better resolution could lead to better simulations of the boundary-layer in areas of complex terrain. I see it is touched on in more detail in

a later paragraph. Should the order of the paragraphs be switched? 17. Page 10, line 1-2. I don't quite get the sentence "... however worsen the representation...". How can you say that the representation is worse? Shouldn't the higher resolution still be a benefit to the simulations? What data is being used to make this argument? Is it just inferred from the changes in temperature bias?Ăă 18. Page 11, line 20-21. What is meant by "both collections"? 19. Page 11, line 29-32. I am not sure that I get the point of this paragraph. As it is written it seems almost circular to me.Ăă 20. Page 13, line 1. Is "pattern" missing after spatial? 21. Page 13, line 17. The test states "...increases night-time biases ..." Is this really fair to say? What is the meaning of the MODIS LST when clouds are present? Shouldn't the cloudy cases be left out of the analysis completely? 22. Page 14, line 6. What is meant by "in runs"? 23. Page 14, line 13-14. Is it fair to say "under representation"? Do you have a measure of the bare-soil fraction?Ăă Could you say sensitivity? 24. Page 15, line 1. The text states "...represents the available energy..." Is the data shown in Figure 7 only for cloud free conditions? 25. Page 15, line 15-19. The text describes biases in the latent heat flux. Could the results also be explained in the context of soil moisture? Could the soil be too moist or the atmosphere too dry (or some combination of both)? Would this have an impact on your results? 26. Page 15, line 31. The text about the location of the radiometers could be rephrased. I assume that the radiometers are mounted above the canopy top or in a fashion that gives a clear view of the sky. 27. Page 17, line 3-5. I commented on this earlier, but I think that one needs to be careful about the use of "better" and "worsen" describing the surface fractions when there isn't a data set that can be used to evaluate the values used in the model.Ăă 28. Figure 1. What does O-B mean? 29. Figure 2. Could the caption be augmented to state the meaning of the shading for the red and blue curves? It would be helpful to indicate the relevant years somewhere on the panels.Ăă 30. Figure 4 (and others). In a number of the figures, the authors may want to consider more descriptive headings on some of the plots. That can orient readers without having to read all of the caption, and I often find it helpful when flipping between the text and figures.Ăă

---

## Author Comment (AC1) · 27 Feb 2019

**Response to Reviewers**

**Anonymous Referee 1**

The manuscript provides an assessment of land surface temperature (LST) simulated with different configurations of the UK Met Office Unified model. The exercise is made for a small area in the US (Arizona) taking advantage of simulations and data gathered for a particular experiment (SALSTICE), which was focused on model LST bias with respect to IASI retrievals. Model LST simulations are compared with IASI and MODIS products, as well as with in situ estimates. Model net radiation, turbulent heat fluxes at the surface and ground flux are also compared with ground observations.

The manuscript is very well written and the subject is of interest, given the limitation in the assimilation of radiances sensitive to lower troposphere over land due to the large model skin temperature biases. However, it is difficult to draw solid conclusions when different model configurations (in terms of dynamics, resolution, approach to bias correction, surface parameters) are not run for a common period. I suggest the article to be accepted subject to revisions in line with my comments below.

We thank Reviewer 1 for carefully reading our paper and providing recommendations and comments on how to improve the manuscript. We believe that the advice in this review is very useful, and contributes to a substantial improvement of the article.

1) On local estimates of LST (section 2.2.2): I fully agree with the need to account for the uncertainty in local emissivity for the LST ground estimates. From the description provided in this section, it seems you do not correct the surface leaving radiance for the downward radiation that is reflected by the surface. This may be the same order of emissivity uncertainty for the 8-14 micro-m band. Please check and modify the data and model versus in situ comparisons in the manuscript as needed.
Thank you for this suggestion, we now apply a further correction to the IRT in-situ data which accounts for the 8-14 µm downwelling longwave radiation according to Eq. (1).

[revised manuscript text omitted]

4) Lines 10-15 (page 8): Please include a short justification for the use of different ZoH/ZoM ratios in the global and limited area model versions.
The $z_{OH}/z_{OM}$ ratio was revised between GA/L3.1 and GA/L6.1 in order improve both land surface temperature and near surface air temperatures in desert regions. The revised $z_{OH}/z_{OM}$ ratio was adopted in the US2.2 (and other LAMs) from 2013, whilst GA/L6.1 was adopted for operational use in July 2014. We have included this text in the manuscript.

5) On the overall analysis of model simulations: As referred above in my general comment, the results of different model configurations correspond to the same period of the year (May), but for model runs performed for different years. You must ensure that when comparing these results, they are not affected by inter-annual variability. In order words, please show that the conditions observed in each May of the 2013-2018 period do not deviate greatly from the average. In the case they do, please check how that may have affected your results. This is relevant, since a dryer or rainier than usual year may lead to a significant change in vegetation cover (and therefore in surface parameters such as surface albedo and emissivity, and even ZoM), soil moisture availability (and likely in the partition between latent and sensible heat fluxes), which will certainly impact your model performance.

Thank you for this suggestion to investigate if the different case periods are affected by inter-annual variability. We have approached this by examining the in situ soil moisture measurements and using the soil moisture anomaly product from Climate Prediction Center for the six year evaluation period.

We have included in the manuscript "Variability of surface temperatures could arise due to variability in cloud cover or soil moisture. In this study we consider only clear sky situations; both the model and observational datasets have been screened to remove cloud contamination, which suggests that soil moisture variability between the analysis years could be a factor for investigation. Point scale measurements of volumetric soil moisture at the eddy-covariance sites are made at depths of 5 cm and 15 cm. A six year multi-year mean soil moisture for each site and at each soil depth has been calculated, and used to calculate a soil moisture anomaly. At both sites, the volumetric soil moisture in May is less than 0.05 kg m$^{-2}$ (0.10 kg m$^{-2}$) at 5 cm (15 cm) for all years in the evaluation. The in situ volumetric soil moisture measurements suggest that the moisture levels were almost always exhausted for each May analysis period and therefore it is unlikely there was sufficient soil moisture to impact on surface temperature variability.

In support of the eddy-covariance measurements, monthly 0.5°×0.5° soil moisture and soil moisture anomaly product from Climate Prediction Center (Fan et al. 2004) were used to assess the larger scale trends in soil moisture in southeastern Arizona. The soil moisture anomaly product indicates that May 2013 and 2014 were anomalously dry (-20 to -40 mm) for an extensive region of the western US, May 2015 had a neutral soil moisture anomaly, May 2016 and 2017 had localised dry regions confined within Arizona, and May 2018 was anomalously dry (-80 mm) for an extensive region of the western US."

Reference added: Fan, Y., and van den Dool, H.: Climate Prediction Center global monthly soil moisture data set at 0.5° resolution for 1948 to present, J. Geophys. Res., 109, D10102, doi:10.1029/2003JD004345, 2004.

6) Last line of page 9 – line 7 of page 10: I'm not sure I follow what is meant here, especially with what respects the degradation in the representation of the grassland fractions. In contrast to the latter, when use a higher resolution land cover, you get better representation for bare soils: is this so? Why? Please clarify (or just rephrase).
Thank you for highlighting the confusion in the text, a similar comment was also raised by Reviewer 2. We have revised this paragraph to compare to modelled bare soil cover with the observation fractions from Scott et al. 2015, and only reference the changes in surface fractions to the four sites and not to the land classification. The revised paragraph is as follows: "The higher resolution ancillaries in the US2.2 improve the surface fractions for the two shrubland sites; the US2.2 increases the bare soil fractional cover which acts to increase the sparsity of the vegetation cover, and improves the model representation of the surface heterogeneity. At the Lucky Hills shrubland site, for example, the bare soil fraction is increased from 0.26 (GA/L3.1) to 0.48 (US2.2_ConfigA-D) and at Santa Rita Mesquite a similar increase from 0.22 (GA/L3.1) to 0.37 (US2.2_ConfigA-D) is reflected. This brings the modelled bare soil cover fractions closer to the observed fractions of 63 % for Lucky Hills Shrubland and 50 % for Santa Rita Mesquite (Scott et al., 2015). However, at the two grassland sites, Kendall Grassland and Santa Rita Grassland, there was a reduction in bare soil fractional cover between GA/L3.1 and US2.2_ConfigA. The lower cover fraction at the grassland sites is maintained in all GA/L6.1_17km configurations. At the Kendall Grassland site, for example, the bare soil fraction is decreased from 0.26 (GA/L3.1) to 0.20 (US2.2_ConfigA-D) and at Santa Rita Grassland a similar decrease from 0.16 (GA/L3.1) to 0.10 (US2.2_ConfigA-D) is reflected. This is in contrast with the observed fractions of 60 % for Kendall Grassland and 45 % for Santa Rita Grassland (Scott et al., 2015)."

7) When discussing the statistics between the various model configurations and MODIS LST products (collections 5 and 6), it would be useful to have an idea how both compare with the in situ estimates (please make sure these are properly estimated, as commented above). You may consider adding a table with a summary of all these, including an average of the in situ (or MODIS) LST per site, which would somehow answer my question above on stable the conditions are among the studied years. This may also help you check if there are years/sites for which MODIS (Aqua or Terra) presents higher biases, and therefore help you analysing your model comparisons with MODIS LST.
Thank you for this suggestion to make a more direct comparison between the model configuration, MODIS LST and the in situ IRT measurement. We have included the in-situ measurements in Figure 3 to enable this comparison, rather than adding a table of the data.

In the text we have included more reference to how the IRT measurements compare with the model and MODIS LST. In section 3.2, paragraph 3 we include: "The IRT measurements support this trend; at Lucky

Hills the bias in reduced from -9.0+3.7 K (GA/L3.1) to -3.3+2.3 K (US2.2_ConfigA), whilst the IRT measurements at Kendall Grasslands only show a 2.2 K improvement in the US2.2_ConfigA compared with GA/L3.1." In section 3.2, paragraph 6 we include: "The IRT measurements located at Lucky Hills support the development of the warm bias (0.6+5.4 K in 2013; 1.4+2.6 K in 2015)".

8) On the assessment of model biases and terrain slope: The impact of slope, especially the x-component) surely differs for Terra (morning overpass) and Aqua (afternoon overpass), if this is essentially related to the LST contrast between slopes facing/hiding from the sun. Maybe this effect is more noticeable in the afternoon, and in that case the "Aqua signature" prevails. In any case, the illumination geometry is obviously very relevant for this, and therefore results should be assessed for the two platforms separately.
Thank you for this suggestion. We have separated the Terra and Aqua platforms for calculating the daytime coefficients of correlation for the x-component and y-component of the orographic slope, and have adjusted Figure 6 accordingly and the caption for Figure 6 "N.B In panel b (c) the collection 6 (dotted) Terra and (dot-dashed) Aqua retrievals are separated for presenting the correlations with the x-component (y-component) of the orographic slope."
Within the manuscript we have revised the text as follows; "The coefficients of correlation between the LST bias and x-component (y-component) of the orographic slope have been calculated for the six year analysis period and are presented in Figure 6b (c). The solar illumination geometry of orography changes as a function of time of day, whilst the remotely sensed LST is a directional variable with each satellite platform (Terra and Aqua) maintaining the same angle with respect to the sun. Each platform measures a similar illumination geometry on each overpass, and therefore the coefficients of correlations are calculated separately for the Terra and Aqua retrievals in Figure 6b and 6c. The night-time coefficients of correlation have a value of +/-0.2 which indicates there is a relationship between the two variables, but it is weak and likely insignificant. For the x-component prior to 2018, the daytime coefficient of correlation was positively correlated with a value of 0.41+0.05 (0.28+0.05) for Aqua (Terra) retrievals; and identifies that regions of cold model LST bias are found on easterly slopes and regions of warm model LST bias are found on westerly slopes. We find a stronger correlation between the x-component of the orographic slope and the LST bias for Aqua compared with Terra, whilst the difference between the two platforms was minimal for the y-component of the orographic slope."

9) Lines 15-16 of page 14 "Our findings suggest that the daytime model LST bias could be minimised by increasing the bare soil cover fraction in the study regions". I don't think you can say this, as you are suggesting that you should change the fraction of bare soil, instead e.g., correcting, e.g., model parameters where the fraction of bare soil is low.
Thank you for this comment, by this we mean that our work has identified that the surface cover ancillary datasets do not adequately represent sparse canopies as the bare soil fractions are too low, and suggest that new developments of ancillary datasets should take this into account. We have revised this sentence as follows; "Our findings suggest that the development of surface cover ancillary datasets for sparse canopies is necessary."

10) The comparisons between model and observed net radiation and surface energy fluxes are only discussed for a single site/year. Although the issue of different models run for different periods could make the discussion difficult, it would be interesting to know how the comparison between simulations and observations evolved as the model land surface temperature changed.
Thank you for this suggestion and we agree that investigating the surface energy balance beyond the 2013 analysis we have presented in this manuscript is important. However, as you indicated it is difficult to draw conclusions for the impact of different model parameters when we are examining different time periods. Rather than interpreting the changes to the surface energy balance with the operational coupled configurations presented in this study, we are doing a follow on study which uses offline/standalone JULES driven with observations from the AmeriFlux network and for a greater number of sites, which will enable us to examine the response of the surface energy balance in greater detail. However this follow on work is beyond the scope of this manuscript, and will form a separate publication.

11) Editorial:

- Abstract: "The diurnal cycle of LST in Global Atmosphere/Land 6.1 (GA/L6.1) showed a significant improvement relative to GA/L3.1": Please be more specific (meaning quantitative) here.
Thank you for this comment. We have revised the sentence to "The diurnal cycle of LST in Global Atmosphere/Land 6.1 (GA/L6.1) showed a significant improvement relative to GA/L3.1 with the cold LST biases reduced to -1.4+2.7 K and -3.6+3.0 K for Terra and Aqua overpasses, respectively."

- lines 5-6 (page 6): Suggest replacing "to give site-specific LST for each site." by "to give site-specific LST.".
Changed.

- Figures 4 and 5: suggest the authors include a short title for each panel (e.g., LST bias – US2.2A), to facilitate their interpretation.
Thank you for this suggestion. We have added a title to each panel in Figure 2, 4, 5 and 7.

- line 23 (page 12): "IASI"
Changed.

- Figure 7: Please ensure each individual scatter-plot has the same range in the y- and x-axes, since we are comparing the same variable (model versus observations). For the same reason, please resize the diagrams so that they are closer to a square, i.e., so that the length in the y-axis corresponding to, say, 10 Wm$^{-2}$ roughly matches the same length for 10 Wm$^{-2}$ in the x-axis.
In Figure 7 we have changed the range of the latent heat flux plot to have the same axis range, and replotted the each subplot so they are square.

- Line 12 page 18: Please rephrase sentence.
We have rephrased the sentence as follows; "With recent advances in supercomputing power, the ability to perform high resolution ensemble forecasting, for example within a research LAM such as the US2.2, is becoming viable. This will provide an opportunity to evaluate the impact of forecast uncertainty on the land surface processes, rather than only for the deterministic forecast as has been carried out in this study."

---

## Author Comment (AC2) · 27 Feb 2019

The manuscript "Evaluating the Met Office Unified Model Global Atmosphere/Land 3.1 (GA/L3.1) and Global Atmosphere/Land 6.1 (GA/L6.1) land surface temperature. Outcomes of the SALSTICE campaign" by Brooke et al. describes an investigation of land surface temperature biases using the Met Office Unified Model. Overall, the results are interesting and show aspects of the errors in simulated land surface temperature for a number of different model configurations, and would be of interest to the scientific community. The simulated temperature biases are related to a number of different parameters in the model. For the most part, the manuscript is well written, although some parts could provide more motivation and be made clearer to the reader. Most of my concerns are relatively minor and should be straight forward to address and the manuscript should be acceptable for publication in Geoscientific Model Development after these concerns are addressed.

We thank reviewer 2 for their comments and we have found the advice very constructive which we have found has the resulted of an overall improvement in the clarity of the manuscript. We have responded to the comments below and corrected or altered the manuscript as follows. Specifically we have revised the discussion around cloud screening of each the datasets, and have performed cloud screening of the IRT data so it is consistent with the cloud screening of the MODIS and model data. We have provided more clarity on the model configurations used in the evaluation.

**Major comments:**
1. The title is a bit misleading, given that it only mentions two model configurations while limited area models are also applied in the study. In addition, the title mentions SALSTICE, but it is not clear to me if any SALSTICE data is included. Was the deployment of the eddy-covariance systems considered part of that study, or was SALSTICE only the airborne deployment that is not used at all?
Thank you for this suggestion, we have revised the title of the manuscript to, "Evaluating the Met Office Unified Model land surface temperature in Global Atmosphere/Land 3.1 (GA/L3.1), Global Atmosphere/Land 6.1 (GA/L6.1) and Limited Area 2.2km configurations."
The deployment of the eddy-covariance systems was not considered part of the SALSTICE campaign; the measurements used in the manuscript are from the Ameriflux network. We have removed the reference to SALSTICE in the title.

2. The application of cloud screening applied in the study needs to be described better. It is mentioned in a couple of places, but I think that it has an important impact on the interpretation of the satellite derived LSTs and should be presented in a consistent way, perhaps in the Methodology section.
Thank you for this suggestion, we recognise the discussion of cloud screening was disjointed. We have revised the text around the application of cloud screening to the IRT data (section 2.2), MODIS cloud screening (section 2.3), and the model data (section 2.4), and have included a description of the cloud screening applied to each dataset in the relevant section as follows:

In section 2.2 (page 6) we add "Cloud screening of the IRT data has been performed using coincident observations of downwelling shortwave as no direct measurement of cloud cover is made at the two AmeriFlux sites. The theoretical clear skies downwelling shortwave for each site has been calculated and compared with the measured downwelling shortwave; times where there is a suppression in the observed downwelling shortwave compared with the theoretical calculation has been attributed to the presence of cloud. It was found that on average (for both sites and for the six analysis years) the IRT data was 0.45 K warmer when applying cloud screening which equates to a -0.45 K larger cold model bias. Cloud screening of the IRT data had a smaller impact in May 2013 and May 2018 with a -0.2 K colder model bias when compared with not accounting for cloud, and the largest impact was found for May 2015 and May 2016 contributing to a -0.7 K colder model bias."

In section 2.3 (page 7) we revise the original sentence to "Cloud screening of the MODIS data has been applied; data which was flagged by the MODIS quality algorithm as contaminated by cloud has been removed from the analysis."

We have moved the discussion of cloud screening of the UM data, which was originally in section 2.3 into section 2.4 (page 9). The revised text is as follows "Model cloud-clearing has been performed for all model configurations based on a threshold of total cloud fraction greater than 0.1 for each model grid box. In cases where the combination of model and MODIS cloud clearing resulted in a fraction of the domain contained less than 10 % of data, the comparison was excluded from the analysis as this was taken to indicate cloud in the region that could affect the measurements."

3. A little more background is needed to help the reader understand the need for all of the model configurations that are presented. I believe that one reason is related to changes in model configurations with time, while other differences are related to parameter values.

Thank you for this suggestion to provide more clarity about the different model configurations used in this study. As way of an introduction to Section 2.4 (page 7) on the UM configurations we include more of an explanation about the Met Office operational model development cycle to help explain why parameters differ over time. We revise the text as follows: "The operational models at the Met Office are continually monitored and developed in order to minimise systematic model biases and to improve forecasts. The changes in all model configurations evaluated in this study are part of the operational model development cycle. Understanding how the model configuration changes impact on surface temperatures in the development cycle, for the purpose of assessing where any advances in the assimilation of greater volumes of hyperspectral satellite sounding data, is an important evaluation."

4. The authors should add a few notes regarding some of the calculations. For example, how is the bias computed? Is the correlation coefficient Person's correlation coefficient or something different?

Thank you for this suggestion, and we apologise that a discussion of these calculations were missing in the original manuscript. With reference to the calculation of the model bias, we include a short discussion (section 1.0, page 3): "In this study, we consider the term model bias to be a model error which is systematic rather than random, and refer to the bias as being the model background-minus-observed (B-O), i.e. where the UM, on average, under- or overestimates a quantity relative to an observed state. The study evaluates statistics of the model background-minus-observed (B-O) residuals for a range of model configurations."

With reference to the calculation of the correlation coefficient, we include (section 3.3, page 14): "A linear least-squares regression is performed between the LST biases and the modelled orography (and surface fractional cover) and apply a Pearson product-moment correlation coefficient to measure the strength and direction of the linear relationship between two variables."

**Minor comments:**
1. Page 1, line 12: Should "greater than 2 K", be "greater than 2 K in magnitude"?
Changed.

2. Page 1, line 13: This is related to my major comment 3 and minor comment 13. A number of different model configurations are used in the study, and it is hard to see the reason why in the abstract. If there is space (given the word limit of the abstract) some reasons for application of different model configurations would be helpful.
Thank you for this suggestion, and we have included an extra sentence to the abstract as follows "a range of UM configurations were assessed with different model resolution, land surface cover datasets and bare soil parameterisations."

3. Page 1, line 18. Please define "Terra" on first usage.
Changed.

4. Page 2, line 7-9. The sentence describing the IASI was confusing as written. It seems to imply that the data is never assimilated, but other part of the manuscript seem to describe that these observations are not used only when the errors are large.
Thank you for this comment. This sentence has now been revised to be explicit that it is IASI window channels and lower-tropospheric (below 400 hPa) sounding channels which are not assimilated specifically for land surfaces and daytime periods. The revised sentence is as follows "At the Met Office, IASI (Infrared Atmospheric Sounding Interferometer) surface-sensitive channels, including window channels and lower-tropospheric (below 400 hPa) sounding channels, are rejected during assimilation windows for observations over land surfaces and during daytime periods (Pavelin and Candy, 2014).

5. Page 2 lines 10-17. Could this paragraph be adjusted for those that are not completely up-to-date with the UM and other models used by the Met Office? The second sentence says that LST is not assimilated into the UM, but the next sentence talks about LSTs being applied in the Met Office operational model.
Thank you for pointing out the confusion in this paragraph. We revise the paragraph as follows: "Recently, research trials have been completed at the Met Office which use night-time LST from the European Space Agency GlobTemperature LSTs project (Ghent et al., 2016) in the land data assimilation system; the study demonstrated improvements in near surface air temperature forecasts and soil temperatures (Candy et al.,

2017). The required LST uncertainty for assimilation within the Met Office operational assimilation scheme is less than 2 K in magnitude, and Candy et al., (2017) highlights the large errors in daytime LST which must be overcome in order to further advance NWP data assimilation. Currently, as LSTs are not assimilated into the operational UM, they provide an independent source of data for assessing the performance of the land surface model's surface exchange and the boundary layer schemes (Edwards, 2010)."

6. Page 2, line 18. What is meant by "background" in this context?
Thank you for this comment; we have included a sentence in the first paragraph to provide clarity for the term background and added an additional reference as follows. "The model background refers to a short-range model forecast; each data assimilation cycle uses newly received observations to update the model background in order to produce a model analysis (Rabier et al., 2005)."

7. Page 2, line 31-33. The surface albedo plays an important role in the surface energy budget. Should albedo also be mentioned in this paragraph?
Absolutely, the surface albedo plays an important role and we include an additional sentence in the manuscript "The surface albedo describes the fraction of incident solar radiation reflected by a surface and is an important surface property in controlling the available energy."

8. Page 4, line 4-6. Why are two different time periods used in this study? I recognize that it is, at least in part, due to the timing of the SALSTICE study. Are the eddy-covariance measurements only available for the shorter time?
The main reason for the shorter time period in 2013 is so that the analysis is coincident with the analysis performed for the SALSTICE campaign data used in a different study. The eddy-covariance measurements are available for a longer time period.

9. Page 4, line 31-33. Could there also be errors associated with the representativeness of the soil heat flux? You mention this later, but it would also fit here.
Thank you for this comment, we have expanded on potential errors with the soil heat flux measurements, and include the addition of two extra sentences as follows. "Additionally, soil heat flux plates buried in the soil can introduce measurement biases due to difference in conductivity between the measurement plates and the surrounding soil (Gentine et al., 2012). Finally, the ground heat fluxes are point measurements and as such do not represent the variability of fluxes across the fetch/sensing area in the same manner associated with the eddy-covariance measurements."

10. Page 5, line 4. Is there any need to consider clouds in the IRT measurements in order to ensure that they are consistent with the satellite observations? I see something mentioned in section 2.3, but should it also be mentioned here?
Thank you for this suggestion and we have now revised our methodology to incorporate cloud screening of the IRT measurements to be consistent with the satellite observations and the model data. Unfortunately there is no direct measurement of cloud cover from the AmeriFlux sites to be able to perform cloud screening, so as an alternative we have calculated the theoretical clear skies downwelling shortwave at each site with the IRT measurements (Kendall Grassland and Lucky Hills). We have then used the theoretical clear skies SWD to compare with the observed SWD and identify times where the observed SWD in suppressed we attribute this to the presence of cloud. This methodology has been applied to all 6 years of data in the study.

In the manuscript we include the following paragraph "Cloud screening of the IRT data has been performed using coincident observations of downwelling shortwave as no direct measurement of cloud cover is made at the two AmeriFlux sites. The theoretical clear skies downwelling shortwave for each site has been calculated and compared with the measured downwelling shortwave; times where there is a suppression in the observed downwelling shortwave compared with the theoretical calculation has been attributed to the presence of cloud. It was found that on average (for both sites and for the six analysis years) the IRT data was 0.45 K warmer when applying cloud screening which equates to a -0.45 K larger cold model bias. Cloud screening of the IRT data had a smaller impact in May 2013 and May 2018 with a -0.2 K colder model bias when compared with not accounting for cloud, and the largest impact was found for May 2015 and May 2016 contributing to a -0.7 K colder model bias."

In line with comments from Referee 1, who has asked used to account for the reflected downwelling longwave, the IRT data presented in the manuscript has been revised to account for both the downwelling longwave and for cloud screening. The text and figures have been revised accordingly.

11. Page 5, line 18-22. Is there a reason why you would expect the night-time values to be unreliable, but not the daytime values?
Unfortunately we have not been able to attribute a cause to the unreliable IRT night-time temperatures.

12. Page 6, line 8-9. I can understand why you wanted to include information about the cloud clearing here, but would it make more sense in the modeling section?
Thank you for this suggestion, we have split this paragraph with the MODIS cloud clearing description in Section 2.3, and moved the description of model cloud clearing into Section 2.4. For clarity we revise the paragraph in Section 2.4 to "Model cloud-clearing has been performed for all model configurations based on a threshold of total cloud fraction greater than 0.1 for each model grid box. In cases where the combination of model and MODIS cloud clearing resulted in a fraction of the domain contained less than 10 % of data, the comparison was excluded from the analysis as this was taken to indicate cloud in the region that could affect the measurements."

13. Page 6, line 19. Section 2.4 could be improved with additional background information regarding the selection of the various model configurations. Why are so many model configurations used? Why are both global and regional models used? I believe that one reason are new versions of the operational model. Table 1 helps, but probably isn't sufficient.
Thank you for this suggestion to provide more clarity about the different model configurations. We refer our response to 'major comment 3' which provides details of the extra additions to the manuscript to explain the different model configurations

14. Page 8, line 19-21. Is the sign convention the same in Figure 1 and the text? Based on the figure it looks like nearly all of the biases are positive, not negative.
We apologise for the confusion between the text and Figure 1. Figure 1 presents the surface temperature biases as observations-model background (O-B) whilst the text describes the biases in the context of model background-observations for consistency with the remainder of the manuscript which present the biases as model-observations. We have clarified the manuscript so that it is explicit that the Figure refers to O-B and the test refers to B-O as follows;
"The surface temperature biases (observations–model background, O-B) for the southern part of the North American continent are presented in Figure 1 for IASI 1D-VAR retrievals compared with two UM global configurations, GA/L3.1 (May 2013) and GA/L6.1_17km_static (May 2015). The IASI 1D-Var retrievals have a spatial resolution of 11 km and have been regridded to a half degree global resolution. In terms of model background-observations (B-O) surface temperature biases, it can be seen that GA/L3.1-IASI 1D-VAR gives rise to an east-west spatial divide in the magnitude of LST biases with LST cold biases in excess of -10 K in the south-west US, western Mexico and extend east into the Great Plains. Moderate cold LST biases extend into the northern US with biases in the range of -4 to -6 K. The North American mean bias is reduced in GA/L6.1_17km_static-IASI 1D-VAR compared with GA/L3.1-IASI 1D-VAR, although regional biases such as the south-west US are still prominent."

15. Page 9, line 11-14. I agree that the field of view of the IRTs is much smaller than the size of the model grid cell, yet the color shading still shows good agreement between the simulations and the IRT.
Thank you for this comment. We wanted to recognise that differences in the scale of the IRT measurements and the size of the model grid-box, however as you indicate it is also important it recognise the agreement between the IRT and model (GA/L6.1_17km_static) in 2015. We include an additional sentence "As the model configurations have grid squares that are many orders of magnitude larger than this, the IRT-measured LST greatly under sample the variability within the model grid square, however despite this Figure 2c and Figure 2d demonstrate good agreement in the representation of the daytime diurnal cycle."

16. Page 9, line 21-22. I agree that high resolution datasets are likely important, but could other factors also lead the improved performance at higher resolution? For example, better resolution could lead to better simulations of the boundary-layer in areas of complex terrain. I see it is touched on in more detail in a later paragraph. Should the order of the paragraphs be switched?
Thank you for this suggestion, and on re-reading this section we have changed the order in paragraph two and four in improve the flow of the discussion.

17. Page 10, line 1-2. I don't quite get the sentence ". . . however worsen the representation. . .". How can you say that the representation is worse? Shouldn't the higher resolution still be a benefit to the simulations? What data is being used to make this argument? Is it just inferred from the changes in temperature bias?

Thank you for this comment. The representation of the vegetation and bare soil fractions was with reference to the observed fractions from Scott et al., (2015) and not with reference to any changes in the surface temperature bias. We have revised this paragraph to compare to modelled bare soil cover with the observation fractions from Scott et al. 2015, and only reference the changes in surface fractions to the four sites and not to the land classification. The revised paragraph is as follows: "The higher resolution ancillaries in the US2.2 improve the surface fractions for the two shrubland sites; the US2.2 increases the bare soil fractional cover which acts to increase the sparsity of the vegetation cover, and improves the model representation of the surface heterogeneity. At the Lucky Hills shrubland site, for example, the bare soil fraction is increased from 0.26 (GA/L3.1) to 0.48 (US2.2_ConfigA-D) and at Santa Rita Mesquite a similar increase from 0.22 (GA/L3.1) to 0.37 (US2.2_ConfigA-D) is reflected. This brings the modelled bare soil cover fractions closer to the observed fractions of 63 % for Lucky Hills Shrubland and 50 % for Santa Rita Mesquite (Scott et al., 2015). However, at the two grassland sites, Kendall Grassland and Santa Rita Grassland, there was a reduction in bare soil fractional cover between GA/L3.1 and US2.2_ConfigA. The lower cover fraction at the grassland sites is maintained in all GA/L6.1_17km configurations. At the Kendall Grassland site, for example, the bare soil fraction is decreased from 0.26 (GA/L3.1) to 0.20 (US2.2_ConfigA-D) and at Santa Rita Grassland a similar decrease from 0.16 (GA/L3.1) to 0.10 (US2.2_ConfigA-D) is reflected. This is in contrast with the observed fractions of 60 % for Kendall Grassland and 45 % for Santa Rita Grassland (Scott et al., 2015)."

18. Page 11, line 20-21. What is meant by "both collections"?
We have revised the text for clarity. The sentence now reads "it was felt to be important to evaluate MODIS C5 and MODIS C6 in order to access the impact on the magnitude of the model biases."

19. Page 11, line 29-32. I am not sure that I get the point of this paragraph. As it is written it seems almost circular to me.
Thank you for this comment, we have removed this paragraph.

20. Page 13, line 1. Is "pattern" missing after spatial?
Added to text.

21. Page 13, line 17. The text states "...increases night-time biases . . ." Is this really fair to say? What is the meaning of the MODIS LST when clouds are present? Shouldn't the cloudy cases be left out of the analysis completely?
All MODIS LST is cloud masked, this section examines the LST bias with and without cloud masking of the model data to examine the magnitude of the bias.

22. Page 14, line 6. What is meant by "in runs"?
Replaces with 'configurations' for clarity

23. Page 14, line 13-14. Is it fair to say "under representation"? Do you have a measure of the bare-soil fraction? Could you say sensitivity?
Thank you for this suggestion, we only have a measure of the bare soil fraction for the Ameriflux sites and not the general region. We have changed 'under representation' to 'sensitivity of the bare soil'.

24. Page 15, line 1. The text states ". . .represents the available energy. . ." Is the data shown in Figure 7 only for cloud free conditions?
The SEB data presented in Figure 7 is not cloud screened. We revise the sentence as follows: "Figure7a presents the net radiation (NR) for all sky conditions which represents the available energy at the surface from radiation."

25. Page 15, line 15-19. The text describes biases in the latent heat flux. Could the results also be explained in the context of soil moisture? Could the soil be too moist or the atmosphere too dry (or some combination of both)? Would this have an impact on your results?
Thank you for this suggestion; examining *in situ* volumetric soil moisture measurements made at depths of 5 cm and 15 cm, it was found that the US2.2_ConfigA soil moisture in the top model level agreed well with the observation at 5 cm. The soil moisture in the second model level was marginally overestimated compared with the 15 cm observations, and could contribute towards the overestimate in the latent heat flux. The latent heat flux is small however compared with the sensible heat flux, and the bias in the sensible heat flux. The relationship between the soil moisture and latent heat flux is complex and dependent on a

number of factors including the vegetation rooting depth, the stomatal conductance of vegetation, hydraulic properties and how these parameters are represented in the model. This is beyond the scope of this study.

26. Page 15, line 31. The text about the location of the radiometers could be rephrased. I assume that the radiometers are mounted above the canopy top or in a fashion that gives a clear view of the sky.
Thank you for this suggestion, we have revised the sentence as follows; "However, an alternative interpretation could be that at the Kendall Grassland site there is shading at location of the ground heat flux plates from vegetation, whilst the net radiometers are mounted above the vegetation canopy and not subject to the effects of shading, which could lead to the lag in the ground heat flux relative to the radiative forcing."

27. Page 17, line 3-5. I commented on this earlier, but I think that one needs to be careful about the use of "better" and "worsen" describing the surface fractions when there isn't a data set that can be used to evaluate the values used in the model.
Thank you for this comment. We have removed this sentence so we are not drawing conclusions about the representation of the land classifications, and only draw conclusions for the representation of the surface fractions for the four sites where there are observations to compare against.

28. Figure 1. What does O-B mean?
Added "(observed-minus-background, O-B)" to Figure 1 caption.

29. Figure 2. Could the caption be augmented to state the meaning of the shading for the red and blue curves? It would be helpful to indicate the relevant years somewhere on the panels.
We have included a description for the meaning of the shading into Figure 2 caption as follows; "The (red shading) is the standard deviation of the IRT measurements and (blue shading) is the standard deviation of the model data." We have included the relevant configuration and year on each panel.

30. Figure 4 (and others). In a number of the figures, the authors may want to consider more descriptive headings on some of the plots. That can orient readers without having to read all of the caption, and I often find it helpful when flipping between the text and figures.
Thank you for this suggestion. We have added a title to each panel in Figure 2, 4, 5 and 7.

---

## Author Response (AR2)

I have reviewed the revised version of the manuscript "Evaluating the Met Office Unified Model land surface temperature in Global Atmosphere/Land 3.1 (GA/L3.1), Global Atmosphere/Land 6.1 (GA/L6.1) and Limited Area 2.2 km configurations" by Brooke et al. Overall, the authors successfully addressed my comments and I believe that the manuscript will be acceptable for publication in Geoscientific Model Development after consideration of a small number minor comments listed below.

We thank reviewer 2 for their comments and we have revised the manuscript as follows.

**Minor Comments:**

Page 3, line 3. Should there be an e.g. before the reference? I am sure that there are many studies that make a similar point

Thank you for this suggestion. We included e.g. prior to the reference.

Page 6, lines 9-10. The sentence states "Hourly downwelling…", but the previous sentence indicates that the Havemann-Taylor Fast Radiative Transfer Code was used—perhaps this is the code that is used in the ECMWF? It would be helpful if the sentence were clarified.

Thank you for this comment, and realise the confusion in the original sentence which we have clarified as follows: "Hourly downwelling longwave radiation is calculated using HT-FRTC based on the ECMWF ERA-Interim (Dee et al., 2011) atmospheric profiles of temperature, specific humidity and ozone mass mixing ratio which are available every 6 hours (00, 06, 12 and 18)."

Page 6, line 17. Did the theoretical calculation account for aerosol loading? I don't think it makes a difference in this application, but the text could mention if aerosol was considered or not.

Thank you for this comment, no the calculation was performed for a clear skies simulation without accounting for aerosol loading. We include the sentence "The downwelling calculation does not account for aerosol loading."

Page 10, lines 3-11. I appreciate that the authors have tried to make it clearer that the O-B is used in the text, while B-O is plotted in the figure. Would it be difficult to replot the figure to match the usage in the body text? Alternatively, a sentence could be added to make it clear that the definition used in the figure is different.

Unfortunately, Figure 1 is not simple to replot as O-B, and as such we include an additional sentence to make it clear of the difference between Figure 1 and manuscript text.

N.B Figure 1 presents the surface temperature bias as O-B, whilst the manuscript text going forward is presented for model-background-minus-observed, B-O.

Page 15, line 23. Rather than "not shown" the text could point to the location in the manuscript where the cloud clearing is described.

Thank you for this suggestion. We have replaced 'not shown' with 'as described in section 2.4'.